# Policy Optimization for Robust Average Cost MDPs

**Zhongchang Sun**
University at Buffalo
zhongcha@buffalo.edu

**Sihong He**
University of Texas at Arlington
sihong.he@uta.edu

**Fei Miao**
University of Connecticut
fei.miao@uconn.edu

**Shaofeng Zou**
Arizona State University
zou@asu.edu

## Abstract

This paper studies first-order policy optimization for robust average cost Markov decision processes (MDPs). Specifically, we focus on ergodic Markov chains. For robust average cost MDPs, the goal is to optimize the worst-case average cost over an uncertainty set of transition kernels. We first develop a sub-gradient of the robust average cost. Based on the sub-gradient, a robust policy mirror descent approach is further proposed. To characterize its iteration complexity, we develop a lower bound on the difference of robust average cost between two policies and further show that the robust average cost satisfies the Polyak-Łojasiewicz (PL)-condition. We then show that with increasing step size, our robust policy mirror descent achieves a linear convergence rate in the optimality gap, and with constant step size, our algorithm converges to an $\epsilon$-optimal policy with an iteration complexity of $\mathcal{O}(1/\epsilon)$. The convergence rate of our algorithm matches with the best convergence rate of policy-based algorithms for robust MDPs. Moreover, our algorithm is the first algorithm that converges to the global optimum with general uncertainty sets for robust average cost MDPs. We provide simulation results to demonstrate the performance of our algorithm.

## 1 Introduction

Markov decision process (MDPs) [34] has been widely used to model agent-environment interactions in sequential decision-making problems. An MDP consists of a set of states, a set of actions, a transition kernel describing the dynamics of the environment, and a cost function of state-action pairs. The agents aim to minimize the cumulative cost obtained over time under a given transition kernel. However, real-world environments often exhibit uncertainties and non-stationarity that challenge the assumptions of traditional RL approaches. When there is a mismatch between the training environment and the real environment, minimizing the cumulative cost under the training environment may lead to poor performance under the real environment.

To address the challenges raised by the model mismatch, the robust MDP was proposed [13, 27], where the transition kernel of the MDP is not fixed but lies in an uncertainty set. The goal of the robust MDP is to optimize the worst-case performance over the uncertainty set of transition kernels. The obtained policy under the robust setting is thus robust to the model mismatch.

Existing works on robust MDPs mainly focus on the discounted cost setting, where the goal is to minimize the worst-case cumulative discounted cost. However, in many real-world applications with long time horizons, such as inventory management in supply chains and applications in communication networks [16], the optimal policies obtained from the discounted cost setting may have poor long-term performance [15].

38th Conference on Neural Information Processing Systems (NeurIPS 2024).

To address these challenges, recent research has shifted focus towards robust average cost MDPs, where the objective is to optimize the worst-case average cost obtained per time step. The average cost MDPs offer several advantages over discounted cost MDPs, including better stability and applicability to infinite-horizon tasks. However, achieving robust and efficient learning in average cost settings remains a significant challenge. Since the cost is discounted exponentially with time in discounted cost MDPs, establishing a contraction only requires a discount factor strictly less than one. Compared with discounted cost MDPs, average cost MDPs depend on the long-term performance of the underlying MDPs as it assigns equal weight to both immediate and future costs. Research on robust average cost MDPs is relatively scarce in the existing literature, with only a few notable studies, such as those by [36, 22, 39, 40, 12]. None of the above-mentioned works focuses on the fundamental characterization of gradient-based algorithms. Therefore, in our paper, we present the first theoretical analysis of the global convergence of policy optimization algorithms in the context of robust average cost MDPs with general uncertainty sets.

## 1.1 Main Contributions

In this paper, for ergodic Markov chains, we propose a policy-based optimization algorithm called robust policy mirror descent to solve the robust average cost MDPs. We further show that with increasing step size, our robust policy mirror descent achieves linear convergence in the optimality gap, and with constant step size, our algorithm converges to an $\epsilon$-optimal policy with iteration complexity of $\mathcal{O}(1/\epsilon)$. Our algorithm is the first policy-based algorithm with global convergence and finite iteration complexity analysis for robust average cost MDPs with general uncertainty sets. In particular, our main contributions are summarized as follows.

**We derive a policy (sub)-gradient for the robust average cost MDPs.** In robust average cost MDPs, the goal is to optimize the worst-case performance, known as the robust average cost, which considers the worst-case over the uncertainty set of transition kernels. However, the robust average cost function typically involves a "max" operator over transition kernels, making it non-differentiable with respect to the policy. Therefore, in this paper, we first develop the Fréchet sub-gradient of the robust average cost with general uncertainty sets, which serves as a foundation of the policy-based algorithm. The robust policy gradient under the discounted setting was derived in [42, 21]. The work [42] considers the specific $R$-contamination uncertainty set, and the derivative of the robust policy gradient in [21] relies on the fact that the discount factor is strictly less than $1$, which can not be extended to the average cost setting.

**We propose a robust policy mirror descent algorithm.** Based on the derivative of the Fréchet sub-gradient, we propose the robust policy mirror descent algorithm. We apply the dynamically weighted divergence to the policy mirror descent so that the policy can be updated for each state separately, which further ensures global convergence.

**We show that our algorithm converges to the global optimum, and we further characterize the iteration complexity.** To show the global convergence, we first prove that the robust average cost satisfies the Polyak-Łojasiewicz (PL) condition [29, 23]. We then prove the global optimality of our algorithm and characterize its iteration complexity. We show that with increasing step size, our robust policy mirror descent achieves linear convergence in the optimality gap, and with constant step size, our algorithm converges to an $\epsilon$-optimal policy with iteration complexity $\mathcal{O}(1/\epsilon)$. For increasing step size, the linear convergence of our algorithm matches with the robust discounted cost setting [21]. In [21], the robust policy mirror descent was shown to converge to the global optimum with iteration complexity $\mathcal{O}(1/\epsilon)$ for discounted cost MDPs when the step size is sufficiently large. Conversely, for non-robust discounted cost MDPs, the policy mirror descent was demonstrated to achieve a $\mathcal{O}(1/\epsilon)$ iteration complexity in [45], representing the state-of-the-art convergence rate for policy mirror descent with a constant step size. Therefore, the convergence rate of our algorithm matches with the performance of the best non-robust counterpart. The convergence analysis of policy-based algorithms for non-robust MDPs relies on the fact that the value function is smooth, which is not the case for the robust average cost function. In this paper, we combine the first-order optimality condition of the policy update and the PL condition to develop a novel proof for the convergence rate of our algorithm.

## 1.2 Related Works

In this section, we discuss works on policy-based approaches for non-robust MDPs, robust discounted cost MDPs and robust average cost MDPs.

**Policy-based approaches for non-robust MDPs.** In the non-robust setting, policy-based algorithms [44, 35, 17, 14, 31, 32] have demonstrated remarkable success across various applications. Recently, the global convergence of the policy-based algorithms were established [4, 5, 1, 24, 20, 19, 49, 9, 48, 45]. For discounted cost MDPs, it was shown in [1] that the projected gradient descent converges to a global optimum with iteration complexity $\mathcal{O}(1/\epsilon^2)$. In both [48] and [45], the authors show that the projected gradient descent converges to a global optimum with a less iteration complexity $\mathcal{O}(1/\epsilon)$. For average cost MDPs, [18] presents a sublinear convergence bound for projected gradient descent, where the bound involves the parameter that characterizes the complexity of the underlying MDP. In all the above works, the convergence analysis depends on the smoothness of the value function. However, since the smoothness may not hold for the robust value function, the methodologies applied in [1, 48, 45, 18] can not be extended to our case. The policy mirror descent with increasing step size was shown to achieve linear convergence in [45]. The policy mirror descent with constant step size was also proved to converge with iteration complexity $\mathcal{O}(1/\epsilon)$ for discounted cost MDPs in [45]. Their proof relies on the performance difference lemma and the fact that the underlying transition kernel doesn't change with time. In this paper, we derive the policy sub-gradient for robust average cost MDPs and design the robust policy mirror descent. We then develop a lower bound on the difference of robust average cost between two policies and further combine it with the first-order optimality condition of the policy update to characterize the iteration complexity of our algorithm.

**Robust discounted cost MDPs.** Robust discounted cost MDPs have been widely studied [13, 27, 41, 3, 11, 47, 21, 46, 28, 33], where the goal is to minimize the worse-case cumulative discounted cost over the uncertainty set of transition kernels. In this section, we introduce works on policy-based algorithms, which are closely related to our work. In [42], the robust MDPs are considered under the $R$-contamination uncertainty set and the robust policy gradient algorithm is designed. It is shown in [38] that the robust policy gradient algorithm converges to the global optimum with iteration complexity $\mathcal{O}(1/\epsilon^3)$. Later in [38], the double-loop robust policy gradient was proposed for general uncertainty sets and was further proved to converge to the global optimum with iteration complexity $\mathcal{O}(1/\epsilon^4)$. The robust policy mirror descent was designed for discounted cost MDPs in [21]. With increasing step size, the robust policy mirror descent converges linearly to the global optimum, while with constant but sufficiently large step size, the algorithm converges to the global optimum with iteration complexity $\mathcal{O}(1/\epsilon)$. In this paper, we study the average cost setting and show that with increasing step size, the robust policy mirror descent achieves linear convergence, and with constant step size, our algorithm converges to the global optimum with iteration complexity $\mathcal{O}(1/\epsilon)$, which matches with the best achievable iteration complexity of policy-based algorithms for robust MDPs. For the case with constant step size, our analysis doesn't require the step size to be sufficiently large.

**Robust average cost MDPs.** Existing literature that focuses on robust average cost MDPs is relatively limited. The robust average cost MDPs were initially explored by [36], where a specific finite interval uncertainty set was considered and the $\mathcal{O}(1/\epsilon)$ convergence rate was achieved. In [22], the robust average cost MDPs were studied under the $l_1$ uncertainty set. However, the approaches in [36, 22] are not applicable for general uncertainty sets. Later in [39, 40, 12], the robust value iteration based algorithms were proposed and the global convergence was proved. In [12], the connection are built between the discounted reward MDPs and average reward MDPs and the existence of Blackwell optimal policies are proved. In [39, 40], the model-based and model-free robust average reward MDPs are studied and the robust relative value iteration algorithms are proposed. However, finding a stopping criterion and characterizing the iteration complexity for robust value iteration based algorithms remain elusive. In this paper, we propose the first policy-based algorithm for robust average cost MDPs. We show that our algorithm converges to the global optimum and we further characterize its iteration complexity. Therefore, our algorithm is the first algorithm with finite iteration complexity analysis for robust average cost MDPs with general uncertainty sets.

**Exponential cost robust MDPs.** For the robust average cost MDPs, when the uncertainty set is defined by the KL-divergence metric, the problem admits a dual formulation, which is the exponential cost robust MDPs. The exponential cost robust MDPs have also been studied in the literature. In [6], the Q-learning and the actor-critic method are described and the asymptotic performance are characterized for risk sensitive robust MDPs. In [7], the value iteration and policy iteration algorithms are also analyzed for risk sensitive MDPs. Recently, in [26], the modified policy iteration is proved to converge to the global optimum for exponential cost risk sensitive MDPs. The policy gradient algorithm for the risk sensitive exponential cost MDPs is studied and the asymptotic convergence

bounds to a stationary point are provided in [25]. In our paper, we study the robust average cost MDPs with general uncertainty sets and characterize the global convergence of our algorithm.

## 2 Preliminaries and Problem Formulation

In this section, we introduce some preliminaries on discounted cost MDPs, average cost MDPs and present our problem formulation.

### 2.1 Discounted Cost MDPs

A discounted cost MDP is defined by the tuple $(\mathcal{S}, \mathcal{A}, \mathsf{P}, r, \gamma)$, where $\mathcal{S}$ denotes the finite state space, $\mathcal{A}$ denotes the finite action space, $\mathsf{P} = \{\mathsf{P}_s^a \in \Delta(\mathcal{S}), s \in \mathcal{S}, a \in \mathcal{A}\}^1$ is the transition kernel, $r : \mathcal{S} \times \mathcal{A} \to [0, 1]$ denotes the cost function, and $\gamma \in [0, 1)$ is the discount factor. Denote by $S$ the number of states and $A$ the number of actions, respectively.

We consider the set of all stationary and randomized policies $\Pi = \{\pi : \mathcal{S} \to \Delta(\mathcal{A})\}$. For each policy $\pi$, it maps from state $s \in \mathcal{S}$ to a distribution over action $a \in \mathcal{A}$. At each state $s$, the agent takes action $a$ with probability $\pi(a|s)$, and the environment transits from state $s$ to state $s'$ according to $\mathsf{P}_s^a$. The discounted value function of a policy $\pi$ starting from an initial state $s$ is defined as

$$V_{\mathsf{P},\gamma}^\pi(s) \triangleq \mathbb{E}_\mathsf{P}\left[\sum_{t=0}^\infty \gamma^t r(s_t, a_t)|s_0 = s, \pi\right], \tag{1}$$

where $\mathbb{E}_\mathsf{P}$ denotes the expectation with respect to the distribution induced by the transition kernel $\mathsf{P}$. To align with conventions in the optimization literature, in this paper, we adopt a minimization formulation. For discounted cost MDPs, the goal is to find a policy $\pi$ that minimizes the discounted value function $V_{\mathsf{P},\gamma}^\pi(s)$ for any initial state $s$.

### 2.2 Average Cost MDPs

Average cost is another fundamental criterion for MDPs. For discounted cost MDPs, the agent penalizes the future cost with discount factor $\gamma$ to demonstrate the preference for the current cost. The average cost MDPs focus on the long-term performance of the underlying MDPs under the steady-state distribution. The average cost MDP can be defined by the tuple $(\mathcal{S}, \mathcal{A}, \mathsf{P}, r)$. For a policy $\pi$, define the average cost under transition kernel $\mathsf{P}$ starting from an initial state $s$ as follows

$$g_\mathsf{P}^\pi(s) \triangleq \lim_{T \to \infty} \mathbb{E}_\mathsf{P}\left[\frac{1}{T}\sum_{t=0}^{T-1} r(s_t, a_t)|s_0 = s, \pi\right]. \tag{2}$$

We also define the relative value function $V_\mathsf{P}^\pi$ and the relative state-action value function $Q_\mathsf{P}^\pi$ for average cost MDPs as follows

$$V_\mathsf{P}^\pi(s) \triangleq \mathbb{E}_\mathsf{P}\left[\sum_{t=0}^\infty \left(r(s_t, a_t) - g_\mathsf{P}^\pi\right)|s_0 = s, \pi\right],$$

$$Q_\mathsf{P}^\pi(s, a) \triangleq \mathbb{E}_\mathsf{P}\left[\sum_{t=0}^\infty \left(r(s_t, a_t) - g_\mathsf{P}^\pi\right)|s_0 = s, a_0 = a, \pi\right]. \tag{3}$$

The relative value function $V_\mathsf{P}^\pi$ and the average cost $g_\mathsf{P}^\pi$ satisfy the following Bellman equation [30]

$$V_\mathsf{P}^\pi(s) = \sum_{a \in \mathcal{A}} \pi(a|s)\left(r(s, a) - g_\mathsf{P}^\pi(s) + \sum_{s' \in \mathcal{S}} \mathsf{P}_{s,s'}^a V_\mathsf{P}^\pi(s')\right), \tag{4}$$

where $\mathsf{P}_{s,s'}^a$ denotes the probability of transiting to state $s'$ when choosing action $a$ at state $s$. Let $d_\mathsf{P}^\pi$ denote the stationary probability induced by the policy $\pi$ and transition kernel $\mathsf{P}$, and it satisfies that $d_\mathsf{P}^\pi \mathsf{P} = d_\mathsf{P}^\pi$. Similar as [18], we consider the projection of the value function onto the subspace orthogonal to the $\mathbf{1}$ vector so that $V_\mathsf{P}^\pi$ and $Q_\mathsf{P}^\pi$ are unique. For average cost MDPs, the goal is to find a policy $\pi$ that minimizes the average cost $g_\mathsf{P}^\pi(s)$ for any initial state $s$.

---

[1] $\Delta(\mathcal{S})$ denotes the probability simplex defined on $\mathcal{S}$.

## 2.3 Robust Average Cost MDPs

For robust MDPs, the transition kernel $\mathsf{P}$ is not fixed but lies in some uncertainty set $\mathcal{P}$. Define the robust average cost MDP by the tuple $(\mathcal{S}, \mathcal{A}, \mathcal{P}, r)$. In this work, we consider $(s, a)$-rectangular uncertainty set:

$$\mathcal{P} = \bigotimes_{s,a} \mathcal{P}_s^a, \ \mathcal{P}_s^a = \{q \in \Delta(\mathcal{S}) : D(q, (\mathsf{P}_0)_s^a) \le R\}, \tag{5}$$

where $\mathsf{P}_0$ is a known nominal transition kernel, $D$ measures the difference between two distributions, e.g., KL divergence, and $R$ is the pre-specified radius of the uncertainty set.

For robust average cost MDPs, the agent aims to optimize the worst-case performance over the uncertainty set $\mathcal{P}$. Define the worst-case average cost as follows

$$g_{\mathcal{P}}^\pi(s) \triangleq \max_{\mathsf{P} \in \mathcal{P}} \lim_{T \to \infty} \mathbb{E}_{\mathsf{P}} \left[ \frac{1}{T} \sum_{t=0}^{T-1} r(s_t, a_t) | s_0 = s, \pi \right]. \tag{6}$$

Similarly, we denote by $V_{\mathcal{P}}^\pi$ and $Q_{\mathcal{P}}^\pi$ the robust relative value function and the robust relative state action value function, respectively. The robust relative value function $V_{\mathcal{P}}^\pi$ and the robust average cost $g_{\mathcal{P}}^\pi$ satisfy the following Bellman equation [39].

$$V_{\mathcal{P}}^\pi(s) = \sum_{a \in \mathcal{A}} \pi(a|s) \big( r(s,a) - g_{\mathcal{P}}^\pi(s) + \max_{\mathsf{P} \in \mathcal{P}} \sum_{s' \in \mathcal{S}} \mathsf{P}_{s,s'}^a V_{\mathcal{P}}^\pi(s') \big). \tag{7}$$

For any policy $\pi$, denote by $d_{\mathcal{P}}^\pi$ the stationary distribution of the state under the worst-case transition kernel of $\pi$. The goal is to find a policy $\pi$ such that the worst-case average cost $g_{\mathcal{P}}^\pi$ is minimized, i.e.,

$$\min_{\pi \in \Pi} g_{\mathcal{P}}^\pi(s), \text{ for any } s \in \mathcal{S}. \tag{8}$$

Denote the optimal policy by $\pi^*$ and the robust average cost of $\pi^*$ by $g_{\mathcal{P}}^*$. In this paper, we state the following assumption to guarantee that the average cost is independent of the initial state, which is widely used in the studies of average cost MDPs [43, 50, 10, 37, 39, 40].

**Assumption 2.1.** For any $\pi \in \Pi$ and $\mathsf{P} \in \mathcal{P}$, the induced Markov chain is ergodic.

## 3 Robust Policy Mirror Descent

In this section, we first derive the robust average cost policy gradient. We then propose the robust policy mirror descent algorithm.

### 3.1 Robust Average Cost Policy Gradient

Since the worst-case average cost $g_{\mathcal{P}}^\pi$ takes "max" over all $\mathsf{P} \in \mathcal{P}$, $g_{\mathcal{P}}^\pi$ might not be differentiable. To address this issue, we introduce the concept of Fréchet sub-gradient. Let $\| \cdot \|$ denote the $L_2$ norm of a vector.

**Definition 3.1.** For any function $f : \mathcal{X} \subseteq \mathbb{R}^N \to \mathbb{R}$, the Fréchet sub-gradient $u \in \mathbb{R}^N$ is a vector that satisfies

$$\liminf_{\substack{\delta \to 0 \\ \delta \neq 0}} \frac{f(x + \delta) - f(x) - \langle u, \delta \rangle}{\|\delta\|} \ge 0. \tag{9}$$

When $f$ is differentiable at $x$, the Fréchet sub-gradient $u$ is the gradient of $f$. In this paper, we consider the direct policy parameterization. We derive the sub-gradient for the robust average cost $g_{\mathcal{P}}^\pi$ in the following lemma.

**Lemma 3.2.** Let $\nabla g_{\mathcal{P}}^\pi(s, a) = d_{\mathcal{P}}^\pi(s) Q_{\mathcal{P}}^\pi(s, a)$. Then $\nabla g_{\mathcal{P}}^\pi$ is the Fréchet sub-gradient of $g_{\mathcal{P}}^\pi$.

Note that the Fréchet sub-gradient has been derived for robust discounted cost MDPs [21], of which the $(s, a)$ entry takes the form $\frac{1}{1-\gamma} d_{\mathcal{P}}^\pi(s) Q_{\mathcal{P}}^\pi(s, a)$. Here, with a little abuse of notation, we use $d_{\mathcal{P}}^\pi$ to denote the visitation distribution of policy $\pi$ under the worst-case transition kernel and use $Q_{\mathcal{P}}^\pi$ to denote the worst-case action value function of policy $\pi$. The Fréchet sub-gradient for robust discounted cost MDPs in [21] can not be extended to the average setting since in [21], the discounted factor $\gamma$ is required to be strictly less than 1. In this paper, we derive the Fréchet sub-gradient of robust average cost MDPs by applying the performance difference lemma for average cost MDPs [8] and the Lipschitz property of the relative action value function [18].

---

**Algorithm 1** Robust Policy Mirror Descent

---

**Input:** step size $\eta_k$, initial policy $\pi_0$
**for** $k = 0, 1, \cdots, K - 1$ **do**
   **for** $s \in \mathcal{S}$ **do**
      Update policy: $\pi_{k+1}(\cdot|s) = \arg\min_{p \in \Delta(\mathcal{A})} \left\{ \eta_k \langle Q_{\mathcal{P}}^{\pi_k}(s, \cdot), p \rangle + D(p, \pi_k(\cdot|s)) \right\}.$
   **end for**
**end for**
**Output:** $\pi_K$

---

### 3.2 Robust Policy Mirror Descent

With Lemma 3.2, we are ready to present our robust policy mirror descent algorithm for average cost MDPs. We assume that for a given policy $\pi$, there exists an oracle that outputs the robust relative state action value function $Q_{\mathcal{P}}^{\pi}$. We denote by $D(\pi(\cdot|s), \pi'(\cdot|s))$ the Bregman divergence between two policies $\pi(\cdot|s)$ and $\pi'(\cdot|s)$. We further define the weighted Bregman divergence function $D_d(\pi, \pi') = \sum_{s \in \mathcal{S}} d(s) D(\pi(\cdot|s), \pi'(\cdot|s))$ for any $d \in \Delta(\mathcal{S})$. We define the following robust policy mirror descent with dynamically weighted divergence

$$\pi_{k+1} = \arg\min_{\pi \in \Pi} \left\{ \eta_k \langle \nabla g_{\mathcal{P}}^{\pi_k}, \pi \rangle + D_{d_{\mathcal{P}}^{\pi_k}}(\pi, \pi_k) \right\}, \tag{10}$$

where $\eta_k$ is the step size. Note that $g_{\mathcal{P}}^{\pi}$ might not be differentiable, thus in our algorithm we replace the gradient of $g_{\mathcal{P}}^{\pi}$ by its Fréchet sub-gradient $\nabla g_{\mathcal{P}}^{\pi}$. By plugging in the sub-gradient formula of $g_{\mathcal{P}}^{\pi}$ in Lemma 3.2, we have that

$$\pi_{k+1} = \arg\min_{\pi \in \Pi} \left\{ \eta_k \sum_{s \in \mathcal{S}} d_{\mathcal{P}}^{\pi_k}(s) \langle Q_{\mathcal{P}}^{\pi_k}(s, \cdot), \pi(\cdot|s) \rangle + D_{d_{\mathcal{P}}^{\pi_k}}(\pi, \pi_k) \right\}$$

$$= \arg\min_{\pi \in \Pi} \left\{ \sum_{s \in \mathcal{S}} \left( \eta_k \langle Q_{\mathcal{P}}^{\pi_k}(s, \cdot), \pi(\cdot|s) \rangle + D(\pi(\cdot|s), \pi_k(\cdot|s)) \right) \right\}, \tag{11}$$

which is equivalent to

$$\pi_{k+1}(\cdot|s) = \arg\min_{p \in \Delta(\mathcal{A})} \left\{ \eta_k \langle Q_{\mathcal{P}}^{\pi_k}(s, \cdot), p \rangle + D(p, \pi_k(\cdot|s)) \right\}, \forall s \in \mathcal{S}. \tag{12}$$

We summarize our algorithm in Algorithm 1.

Note that for the projected policy (sub)-gradient algorithm, the policy is updated as follows

$$\pi_{k+1}(\cdot|s) = \arg\min_{p \in \Delta(\mathcal{A})} \left\{ \eta_k \langle \nabla g_{\mathcal{P}}^{\pi_k}(s, \cdot), p \rangle + \|p - \pi_k(\cdot|s)\|^2 \right\}, \forall s \in \mathcal{S}. \tag{13}$$

In our paper, we set the Bregman divergence $D(\cdot, \cdot)$ to be the squared Euclidean distance. In this case, the difference between our robust policy mirror descent and the projected policy gradient lies in that we replace the policy (sub)-gradient $\nabla g_{\mathcal{P}}^{\pi_k}$ by $Q_{\mathcal{P}}^{\pi_k}$.

In the next section, we show that though the robust average cost $g_{\mathcal{P}}^{\pi}$ might not be differentiable, our robust policy mirror descent achieves linear convergence in the optimality gap with increasing step size, and converges to an $\epsilon$-optimal policy with iteration complexity $\mathcal{O}(1/\epsilon)$ with constant step size.

## 4 Theoretical Results

Before we show the global optimality of the robust policy mirror descent, we first provide some important properties of the robust average cost MDPs.

We first provide a lower bound on the difference of robust average cost between two policies, which is a key step to derive the global optimality.

**Lemma 4.1.** *For any two policies $\pi, \pi'$, we have that $g_{\mathcal{P}}^{\pi} - g_{\mathcal{P}}^{\pi'} \geq \mathbb{E}_{s \sim d_{\mathcal{P}}^{\pi'}} \left[ \langle Q_{\mathcal{P}}^{\pi}(s, \cdot), \pi(\cdot|s) - \pi'(\cdot|s) \rangle \right].$*

Lemma 4.1 is not a straightforward extension of performance difference lemma [8] to the robust setting since the worst-case transition kernels are different for different policies. A similar bound was derived in [21] for robust discounted value function by applying the Bellman equation of robust value function, which is not applicable in our case as the average cost itself does not satisfy the Bellman equation. Therefore, we apply the Bellman equation of the robust relative value function $V_{\mathcal{P}}^{\pi}$ and the robust average cost $g_{\mathcal{P}}^{\pi}$ in (7) and the observation that $g_{\mathcal{P}}^{\pi}$ is independent of the initial state $s$ to obtain Lemma 4.1.

We then show that the robust average cost $g_{\mathcal{P}}^{\pi}$ satisfies the PL-condition in the following lemma.

**Lemma 4.2.** *The suboptimality of any $\pi$ satisfies $g_{\mathcal{P}}^{\pi} - g_{\mathcal{P}}^{*} \leq C_{PL} \max_{\hat{\pi}} \langle \nabla g_{\mathcal{P}}^{\pi}, \pi - \hat{\pi} \rangle$, where* $C_{PL} = \max_{\pi,s} \frac{d_{\mathcal{P}}^{\pi^*}(s)}{d_{\mathcal{P}}^{\pi}(s)}$.

The PL-condition implies that when the subgradient $\nabla g_{\mathcal{P}}^{\pi}$ is small, the policy $\pi$ lies in the small neighborhood of the global optimum.

Our convergence analysis also leverages the following Lipschitz property of the non-robust relative value function $V_{\mathsf{P}}^{\pi}$ [18].

**Lemma 4.3.** *The relative value function $V_{\mathsf{P}}^{\pi}$ is Lipschitz in $\pi$, i.e., there exists a constant $L_{\pi}$ such that $|V_{\mathsf{P}}^{\pi} - V_{\mathsf{P}}^{\pi'}| \leq L_{\pi} \|\pi - \pi'\|$.*

## 4.1 Increasing Step Size

In this section, we show that our algorithm achieves linear convergence rate with increasing step size. We first characterize some properties of our robust policy mirror descent algorithm.

**Lemma 4.4.** *For any $p \in \Pi$ and $s \in \mathcal{S}$, we have that*

$$\eta_k \langle Q_{\mathcal{P}}^{\pi_k}(s, \cdot), \pi_{k+1}(\cdot|s) - p(\cdot|s) \rangle + \|\pi_{k+1}(\cdot|s) - \pi_k(\cdot|s)\|^2$$
$$\leq \|p(\cdot|s) - \pi_k(\cdot|s)\|^2 - \|p(\cdot|s) - \pi_{k+1}(\cdot|s)\|^2. \tag{14}$$

In the following lemma, we establish the convergence property for each iteration of our algorithm.

**Lemma 4.5.** *At each iteration of our algorithm, we have that*

$$g_{\mathcal{P}}^{\pi_{k+1}} - g_{\mathcal{P}}^{*} \leq \frac{M-1}{M}(g_{\mathcal{P}}^{\pi_k} - g_{\mathcal{P}}^{*}) + \frac{1}{M}\mathbb{E}_{s \sim d_{\mathsf{P}_{\pi_k}}^{\pi^*}}\left[\frac{1}{\eta_k}\left\|\pi^*(\cdot|s) - \pi_k(\cdot|s)\right\|^2\right]$$
$$- \frac{1}{M}\mathbb{E}_{s \sim d_{\mathsf{P}_{\pi_k}}^{\pi^*}}\left[\frac{1}{\eta_k}\left\|\pi^*(\cdot|s) - \pi_{k+1}(\cdot|s)\right\|^2\right], \tag{15}$$

*where $M = \sup_{\pi,\mathsf{P} \in \mathcal{P}} \left\|\frac{d_{\mathsf{P}_{\pi}}^{\pi^*}}{d_{\mathsf{P}}^{\pi}}\right\|_{\infty}$.*

We show that with increasing step size, our robust policy mirror descent converges linearly.

**Theorem 4.6.** *Under Assumption 2.1, set the step size $\eta_k \geq \eta_{k-1}\left(1 - \frac{1}{M}\right)^{-1}M$. The robust policy mirror descent satisfies*

$$g_{\mathcal{P}}^{\pi_k} - g_{\mathcal{P}}^{*} \leq \left(1 - \frac{1}{M}\right)^k(g_{\mathcal{P}}^{\pi_0} - g_{\mathcal{P}}^{*}) + \left(1 - \frac{1}{M}\right)^{k-1}\frac{1}{M\eta_0}\mathbb{E}_{s \sim d_{\mathsf{P}_{\pi_0}}^{\pi^*}}\left[\|\pi^*(\cdot|s) - \pi_0(\cdot|s)\|^2\right]. \tag{16}$$

Our analysis in this section mainly leverages the performance difference lemma [8] and the Bregman divergence three-point lemma. The convergence rate for our robust policy mirror descent with increasing step size matches with the best convergence rate of policy-based algorithm for robust MDPs [21]. Our algorithm is the first algorithm that converges to the global optimum with finite iteration complexity for robust average cost MDPs with general uncertainty sets.

## 4.2 Constant Step Size

We proceed to show that with constant step size, our algorithm achieves the global optimum with iteration complexity $\mathcal{O}(1/\epsilon)$.

**Theorem 4.7.** *Under Assumption 2.1, let step size $\eta = \frac{1}{L_\pi}$ for all $k \geq 1$. We have that the each iteration of the robust policy mirror descent satisfies*

$$g_{\mathcal{P}}^{\pi_k} - g_{\mathcal{P}}^* \leq \max\left\{ \frac{4L_\pi}{\omega k}, \left(\frac{\sqrt{2}}{2}\right)^k \left(g_{\mathcal{P}}^{\pi_0} - g_{\mathcal{P}}^*\right)\right\}, \qquad (17)$$

*where $\omega = (2\sqrt{2S}C_{PL})^{-2}$.*

For Algorithm 1, to find an $\epsilon$-optimal policy, Theorem 4.7 shows that the iteration complexity is upper bounded by $\mathcal{O}(1/\epsilon)$. In [21], the robust policy mirror descent was studied for discounted cost MDPs. The iteration complexity $\mathcal{O}(1/\epsilon)$ in [21] can only be achieved with a sufficient large step size $\eta_k = 1/\epsilon$. For the non-robust discounted cost MDPs, the policy mirror descent was shown to enjoy a $\mathcal{O}(1/\epsilon)$ iteration complexity in [45], which is the state-of-the-art convergence rate for policy mirror descent with constant step size. Our robust policy mirror descent with constant step size converges to the global optimum with iteration complexity $\mathcal{O}(1/\epsilon)$, which matches with the best non-robust counterpart [45].

Define the gradient mapping $G_{\frac{1}{\eta}}(\pi_k(\cdot|s)) = \frac{1}{\eta}(\pi_k(\cdot|s) - \pi_{k+1}(\cdot|s))$. Note that if $\pi_k$ is updated exactly by the (sub)-gradient descent, then $G_{\frac{1}{\eta}}(\pi_k(\cdot|s)) = \nabla g_{\mathcal{P}}^\pi(s, \cdot)$. The norm $\|G_{\frac{1}{\eta}}(\pi_k(\cdot|s))\|$ measures the closeness of the current step to the first-order stationary point. Our proof relies on the following key ingredient.

$$g_{\mathcal{P}}^{\pi_k} - g_{\mathcal{P}}^{\pi_{k+1}} \geq \mathbb{E}_{s \sim d_{\mathcal{P}}^{\pi_{k+1}}}\left[\|G_{\frac{1}{\eta}}(\pi_k(\cdot|s))\|^2\right] \geq \omega\eta(g_{\mathcal{P}}^{\pi_{k+1}} - g_{\mathcal{P}}^*)^2. \qquad (18)$$

In [1, 45], similar steps are adopted to derive the convergence rate of the policy gradient descent for non-robust discounted cost MDPs. In their analyses, to obtain (18), the smoothness of the value function is required. However, since the worst-case transition kernel is a function of the policy $\pi$, the robust average cost $g_{\mathcal{P}}^\pi$ might not be smooth. Therefore, the approaches applied in [1, 45] can not be directly extended to robust average cost MDPs. Without the smoothness of the value function, we still prove that (18) holds by leveraging Lemma 4.1 and the following first-order optimality, which is from the optimality condition of the robust policy mirror descent update in (12)

$$\left\langle Q_{\mathcal{P}}^{\pi_k}(s, \cdot) + \frac{1}{\eta_k}(\pi_{k+1}(\cdot|s) - \pi_k(\cdot|s)), p - \pi_{k+1}(\cdot|s)\right\rangle \geq 0, \forall p \in \Delta(\mathcal{A}). \qquad (19)$$

*Remark* 4.8. Since for robust discounted cost MDPs, Lemma 4.1, Lemma 4.2 and Lemma 4.3 also hold, the result in Theorem 4.7 also holds for robust discounted cost MDPs. Therefore, our robust policy mirror descent with constant step size finds an $\epsilon$-optimal policy with iteration complexity $\mathcal{O}(1/\epsilon)$ for robust discounted cost MDPs. Compared with [21], our step size doesn't need to be sufficiently large.

## 5 Simulation Results

In this section, we provide some simulation results to demonstrate the performance of our algorithm. We verify our method on one classical problem: the Garnet problem, and a robotic application problem: the recycling robot problem.

Garnet environments are synthetic benchmarks designed to be used for studying the performance of RL algorithms. The Garnet framework provides a way to create randomly generated MDPs with specified properties, such as the number of states, actions, and transition probabilities. More details can be found in [2].

In the recycling robot problem, a mobile robot powered by a rechargeable battery is tasked with collecting empty soda cans. The robot operates with two battery levels: low and high. It has three possible actions: (1) search for empty cans; (2) remain stationary and wait for someone to bring it a can; or (3) return to its home base to recharge. When the robot's battery is low (high), it has a probability of $\alpha$ ($\beta$) of finding an empty can and maintaining its current battery level. If the robot searches for cans but does not find any, it will deplete its battery completely and must be carried back by humans. For more details, refer to [34]. In this paper, we set $\alpha = 0.9$, $\beta = 0.9$.

We compare our robust policy mirror descent with the non-robust method [45]. We consider Garnet(3, 2), Garnet(5, 2), Garnet(10, 5) and recycling robot problems. Both methods use a uniform random

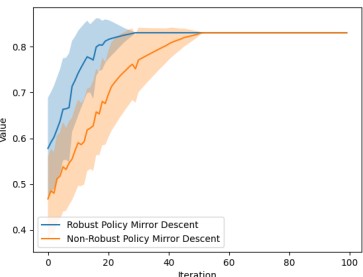

Figure 1: Garnet(3, 2)

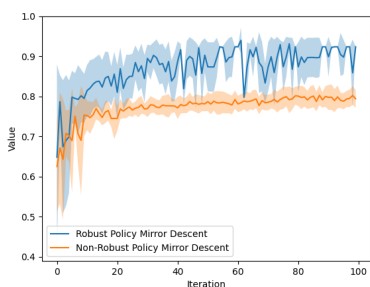

Figure 2: Garnet(5, 2)

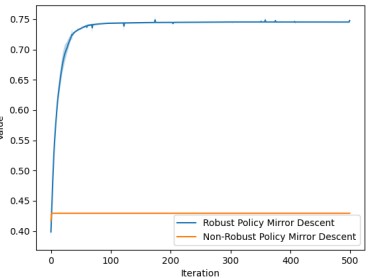

Figure 3: Garnet(10, 5)

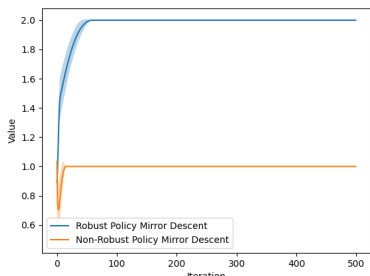

Figure 4: Recycling Robot

policy as the initialized policy. We conducted 5 trials with different random seeds and reported mean and standard deviation of the robust average costs over training episodes. Each training episode contains 2000 training steps. The length of training episodes is respectively 100 and 300 for Garnet and robot problems. We choose the uncertainty set to be the KL divergence uncertainty set. We consider the constant step size and set the step size $\eta = 0.01$, the pre-specified radius of the uncertainty set $R = 0.1$. The host machine used in our experiments is a server configured with AMD Ryzen Threadripper 2990WX 32-core processors and four Quadro RTX 6000 GPUs. All experiments are performed on Python 3.8.

Figure 1 showcases the mean values of the optimal robust average cost values and standard deviations for our robust method and the non-robust baseline over the training episode in Garnet(3, 2). Our robust method converges faster than the non-robust baseline, though these two methods converge to the same value. We speculate that the reason why the two methods converge to similar numbers is because this MDP environment has a small size. Therefore, we further compare our robust method with the non-robust baseline in two larger Garnet problems Garnet(5, 2) and Garnet(10, 5), and the recycling robot problem. Figure 2, 3, and 4 respectively show the mean optimal robust average cost and standard deviations in these three problems. From Figure 2, 3 and 4, we find that our robust method outperforms the non-robust baseline in terms of mean costs in all three problems. The simulation results demonstrate the robustness of our algorithm.

# 6 Conclusion

In this paper, we investigated the policy-based algorithm for robust average cost MDPs. We first introduced the sub-gradient of the robust average cost. Based on the sub-gradient, we proposed the robust policy mirror descent. Our theoretical analysis demonstrates that the proposed algorithm with increasing step size achieves linear convergence rate, and with constant step size, the proposed algorithm finds an $\epsilon$-optimal policy with iteration complexity $\mathcal{O}(1/\epsilon)$, matching the best convergence rate observed in policy mirror descent algorithms for robust MDPs. Moreover, our algorithm is the first algorithm that converges to the global optimum with finite iteration complexity for robust average cost MDPs with general uncertainty sets. Our paper focuses on the model-based setting. In the future, it is of interest to design policy-based model-free algorithms for robust average cost MDPs.

# 7 Acknowledgements

The work of Zhongchang Sun and Shaofeng Zou is supported by the National Science Foundation under Grants CCF-2438429 and ECCS-2438392 (CAREER). The work of Sihong He and Fei Miao is supported by the National Science Foundation under Grants CNS-2047354 (CAREER), and the New England University Transportation Center (NEUTC). Funding for the UTC Program is provided by the Office of Assistant Secretary for Research and Innovation (OST-R) of the United States Department of Transportation. The recommendations of this study are those of the authors and do not represent the views of NEUTC.

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

# A  Proof of Lemma 3.2

In this section, we provide the proof for Lemma 3.2.

Consider polices $\pi, \pi' \in \Pi$. For the non-robust average cost $g_{\mathsf{P}}^\pi$, we have that

$$g_{\mathsf{P}}^{\pi'} - g_{\mathsf{P}}^\pi$$
$$\overset{(a)}{=} \sum_s d_{\mathsf{P}}^\pi(s) \langle Q_{\mathsf{P}}^{\pi'}(s, \cdot), \pi'(\cdot|s) - \pi(\cdot|s) \rangle$$
$$= \sum_s d_{\mathsf{P}}^\pi(s) \langle Q_{\mathsf{P}}^{\pi'}(s, \cdot) + Q_{\mathsf{P}}^\pi(s, \cdot) - Q_{\mathsf{P}}^\pi(s, \cdot), \pi'(\cdot|s) - \pi(\cdot|s) \rangle$$
$$= \sum_s d_{\mathsf{P}}^\pi(s) \langle Q_{\mathsf{P}}^\pi(s, \cdot), \pi'(\cdot|s) - \pi(\cdot|s) \rangle + \sum_s d_{\mathsf{P}}^\pi(s) \langle Q_{\mathsf{P}}^{\pi'}(s, \cdot) - Q_{\mathsf{P}}^\pi(s, \cdot), \pi'(\cdot|s) - \pi(\cdot|s) \rangle, \quad (20)$$

where $(a)$ is from the performance difference lemma [8].

Since the relative value function $V_{\mathsf{P}}^\pi(s)$ is $L_\pi$-Lipschitz in $\pi$ (Lemma 4.3), we have that $Q_{\mathsf{P}}^\pi(s, \cdot)$ is $L_\pi$-Lipschitz in $\pi$. Therefore, we have that $Q_{\mathsf{P}}^{\pi'}(s, \cdot) - Q_{\mathsf{P}}^\pi(s, \cdot) = \mathcal{O}(\|\pi - \pi'\|)$, which further implies that $\sum_s d_{\mathsf{P}}^\pi(s) \langle Q_{\mathsf{P}}^{\pi'}(s, \cdot) - Q_{\mathsf{P}}^\pi(s, \cdot), \pi'(\cdot|s) - \pi(\cdot|s) \rangle = o(\|\pi - \pi'\|)$.

Let $\mathsf{P}_\pi$ denote the worst-case transition kernel for policy $\pi$. For the robust average cost, we have that

$$g_{\mathcal{P}}^{\pi'} - g_{\mathcal{P}}^\pi = g_{\mathsf{P}'_\pi}^{\pi'} - g_{\mathsf{P}_\pi}^\pi \geq g_{\mathsf{P}_\pi}^{\pi'} - g_{\mathsf{P}_\pi}^\pi$$
$$= \sum_s d_{\mathsf{P}}^\pi(s) \langle Q_{\mathsf{P}}^\pi(s, \cdot), \pi'(\cdot|s) - \pi(\cdot|s) \rangle + o(\|\pi - \pi'\|). \quad (21)$$

We then have that

$$\lim_{\pi' \to \pi} \inf_{\pi' \neq \pi} \frac{g_{\mathcal{P}}^{\pi'} - g_{\mathcal{P}}^\pi - \sum_s d_{\mathsf{P}}^\pi(s) \langle Q_{\mathsf{P}}^\pi(s, \cdot), \pi'(\cdot|s) - \pi(\cdot|s) \rangle}{\|\pi - \pi'\|} \geq 0. \quad (22)$$

Therefore, the Fréchet sub-gradient of $g_{\mathcal{P}}^\pi$ is $\nabla g_{\mathcal{P}}^\pi$ with each $(s, a)$-entry $\nabla g_{\mathcal{P}}^\pi(s, a) = d_{\mathcal{P}}^\pi(s) Q_{\mathcal{P}}^\pi(s, a)$.

# B  Proof of Lemma 4.1

For the robust relative value function $V_{\mathcal{P}}^\pi(s)$, we have that

$$V_{\mathcal{P}}^{\pi'}(s) - V_{\mathcal{P}}^\pi(s)$$
$$= \sum_a \pi'(a|s) Q_{\mathcal{P}}^{\pi'}(s, a) - \sum_a \pi(a|s) Q_{\mathcal{P}}^\pi(s, a)$$
$$= \sum_a \left( \pi'(a|s) - \pi(a|s) + \pi(a|s) \right) Q_{\mathcal{P}}^{\pi'}(s, a) - \sum_a \pi(a|s) Q_{\mathcal{P}}^\pi(s, a)$$
$$= \sum_a \left( \pi'(a|s) - \pi(a|s) \right) Q_{\mathcal{P}}^{\pi'}(s, a) - \sum_a \pi(a|s) \left( Q_{\mathcal{P}}^{\pi'}(s, a) - Q_{\mathcal{P}}^\pi(s, a) \right)$$
$$\overset{(a)}{=} \sum_a \Bigg( \left( \pi'(a|s) - \pi(a|s) \right) Q_{\mathcal{P}}^{\pi'}(s, a) + \pi(a|s) \sum_{s'} \mathsf{P}_{\pi'}(s'|s, a) \left( r(s, a) - g_{\mathcal{P}}^{\pi'}(s') + V_{\mathcal{P}}^{\pi'}(s') \right)$$
$$- \pi(a|s) \sum_{s'} \mathsf{P}_\pi(s'|s, a) \left( r(s, a) - g_{\mathcal{P}}^\pi(s') + V_{\mathcal{P}}^\pi(s') \right) \Bigg)$$
$$= \sum_a \Bigg( \left( \pi'(a|s) - \pi(a|s) \right) Q_{\mathcal{P}}^{\pi'}(s, a) + \pi(s|a) \left( g_{\mathcal{P}}^\pi(s) - g_{\mathcal{P}}^{\pi'}(s) \right)$$
$$+ \sum_{s'} \mathsf{P}_{\pi'}(s'|s, a) V_{\mathcal{P}}^{\pi'}(s') - \sum_{s'} \mathsf{P}_\pi(s'|s, a) V_{\mathcal{P}}^\pi(s') \Bigg), \quad (23)$$

where $(a)$ is from the Bellman equation (4). By re-arranging terms and from the fact that $g_{\mathcal{P}}^\pi$ is independent of $s$, we have that

$$g_{\mathcal{P}}^\pi(s) - g_{\mathcal{P}}^{\pi'}(s)$$

$$= \sum_a \left( (\pi'(a|s) - \pi(a|s))Q_{\mathcal{P}}^{\pi'}(s,a) + \pi(s|a)\left( \sum_{s'} \mathsf{P}_{\pi'}(s'|s,a)V_{\mathcal{P}}^{\pi'}(s') - \sum_{s'} \mathsf{P}_\pi(s'|s,a)V_{\mathcal{P}}^{\pi}(s') \right) \right)$$

$$- V_{\mathcal{P}}^{\pi'}(s) + V_{\mathcal{P}}^{\pi}(s)$$

$$\overset{(a)}{\geq} \sum_a \left( \pi'(a|s) - \pi(a|s) \right)Q_{\mathcal{P}}^{\pi'}(s,a) - \sum_a \pi(a|s)\left( \sum_{s'} \mathsf{P}_\pi(s'|s,a)V_{\mathcal{P}}^{\pi'}(s') - \sum_{s'} \mathsf{P}_\pi(s'|s,a)V_{\mathcal{P}}^{\pi}(s') \right)$$

$$- V_{\mathcal{P}}^{\pi'}(s) + V_{\mathcal{P}}^{\pi}(s), \tag{24}$$

where $(a)$ is due to the fact that $\mathsf{P}_\pi$ is the worst-case transition kernel for $\pi$. Since the LHS of (24) is independent of $s$, the RHS of (24) doesn't depend on $s$ either. We then take the weighted sum of the RHS with respect to $d_{\mathsf{P}_\pi}^\pi$:

$$g_{\mathcal{P}}^\pi(s) - g_{\mathcal{P}}^{\pi'}(s)$$
$$\geq \sum_s \sum_a d_{\mathsf{P}_\pi}^\pi(s)\left( \pi'(a|s) - \pi(a|s) \right)Q_{\mathcal{P}}^{\pi'}(s,a) + \sum_s \sum_a d_{\mathsf{P}_\pi}^\pi(s)\pi(a|s)\sum_{s'} \mathsf{P}_\pi(s'|s,a)V_{\mathcal{P}}^{\pi'}(s')$$
$$- \sum_s \sum_a d_{\mathsf{P}_\pi}^\pi(s)\pi(a|s)\sum_{s'} \mathsf{P}_\pi(s'|s,a)V_{\mathcal{P}}^{\pi}(s') - \sum_s d_{\mathsf{P}_\pi}^\pi(s)V_{\mathcal{P}}^{\pi'}(s) + \sum_s d_{\mathsf{P}_\pi}^\pi(s)V_{\mathcal{P}}^{\pi}(s). \tag{25}$$

From the definition of the stationary distribution, we have that $\sum_s \sum_a d_{\mathsf{P}_\pi}^\pi(s)\pi(a|s)\mathsf{P}_\pi(s'|s,a) = d_{\mathsf{P}_\pi}^\pi(s')$. Therefore,

$$g_{\mathcal{P}}^\pi(s) - g_{\mathcal{P}}^{\pi'}(s)$$
$$\geq \mathbb{E}_{s \sim d_{\mathsf{P}_\pi}^\pi}\left[ \langle Q_{\mathcal{P}}^{\pi'}(s,\cdot), \pi'(\cdot|s) - \pi(\cdot|s) \rangle \right] + \sum_{s'} d_{\mathsf{P}_\pi}^\pi(s')V_{\mathcal{P}}^{\pi'}(s') - \sum_{s'} d_{\mathsf{P}_\pi}^\pi(s')V_{\mathcal{P}}^{\pi}(s')$$
$$- \sum_s d_{\mathsf{P}_\pi}^\pi(s)V_{\mathcal{P}}^{\pi'}(s) + \sum_s d_{\mathsf{P}_\pi}^\pi(s)V_{\mathcal{P}}^{\pi}(s)$$
$$= \mathbb{E}_{s \sim d_{\mathsf{P}_\pi}^\pi}\left[ \langle Q_{\mathcal{P}}^{\pi'}(s,\cdot), \pi'(\cdot|s) - \pi(\cdot|s) \rangle \right]. \tag{26}$$

This completes the proof.

## C  Proof of Lemma 4.2

From the performance difference lemma [8], we have that

$$g_{\mathcal{P}}^\pi - g_{\mathcal{P}}^*$$
$$= g_{\mathsf{P}_\pi}^\pi - g_{\mathsf{P}_{\pi^*}}^{\pi^*}$$
$$\leq g_{\mathsf{P}_\pi}^\pi - g_{\mathsf{P}_\pi}^{\pi^*}$$
$$= \sum_s d_{\mathsf{P}_\pi}^{\pi^*}(s)\langle Q_{\mathsf{P}_\pi}^\pi(s,\cdot), \pi(\cdot|s) - \pi^*(\cdot|s) \rangle$$
$$\leq \max_{\hat{\pi}} \sum_s d_{\mathsf{P}_\pi}^{\pi^*}(s)\langle Q_{\mathsf{P}_\pi}^\pi(s,\cdot), \pi(\cdot|s) - \hat{\pi}(\cdot|s) \rangle$$
$$= \sum_s \frac{d_{\mathsf{P}_\pi}^{\pi^*}(s)}{d_{\mathsf{P}_\pi}^\pi(s)} d_{\mathsf{P}_\pi}^\pi(s) \max_{\hat{\pi}}\langle Q_{\mathsf{P}_\pi}^\pi(s,\cdot), \pi(\cdot|s) - \hat{\pi}(\cdot|s) \rangle. \tag{27}$$

Note that $d_{\mathsf{P}_\pi}^\pi(s) \max_{\hat{\pi}}\langle Q_{\mathsf{P}_\pi}^\pi(s,\cdot), \pi(\cdot|s) - \hat{\pi}(\cdot|s) \rangle \geq 0$ since $\langle Q_{\mathsf{P}_\pi}^\pi(s,\cdot), \pi(\cdot|s) - \pi(\cdot|s) \rangle = 0$. We then have that

$$g_{\mathcal{P}}^\pi - g_{\mathcal{P}}^*$$
$$\leq \sum_s \left( \max_{\pi,s} \frac{d_{\mathsf{P}_\pi}^{\pi^*}(s)}{d_{\mathsf{P}_\pi}^\pi(s)} \right) d_{\mathsf{P}_\pi}^\pi(s) \max_{\hat{\pi}}\langle Q_{\mathsf{P}_\pi}^\pi, \pi(\cdot|s) - \hat{\pi}(\cdot|s) \rangle$$
$$= C_{PL} \max_{\hat{\pi}} \sum_s \langle Q_{\mathsf{P}_\pi}^\pi, \pi(\cdot|s) - \hat{\pi}(\cdot|s) \rangle$$

$$= C_{PL} \max_{\hat{\pi}} \langle \nabla g_{\mathcal{P}}^{\pi}, \pi - \hat{\pi} \rangle, \tag{28}$$

where $C_{PL} = \max_{\pi, s} \frac{d_{\mathcal{P}}^{\pi^*}(s)}{d_{\mathcal{P}}^{\pi}(s)}$ and the last step is from Lemma 3.2.

## D  Proof of Lemma 4.4

For the update of our algorithm

$$\pi_{k+1}(\cdot|s) = \arg\min_{p \in \Delta(\mathcal{A})} \left\{ \eta_k \langle Q_{\mathcal{P}}^{\pi_k}(s, \cdot), p \rangle + D(p, \pi_k(\cdot|s)) \right\}, \forall s \in \mathcal{S}, \tag{29}$$

the following first-order optimality condition holds for any $p \in \Pi$ and $s$

$$\langle \eta_k Q_{\mathcal{P}}^{\pi_k}(s, \cdot) + \left( \pi_{k+1}(\cdot|s) - \pi_k(\cdot|s) \right), p(\cdot|s) - \pi_{k+1}(\cdot|s) \rangle \geq 0. \tag{30}$$

From the three-point Lemma of the Bregman divergence, we have that

$$\langle \pi_{k+1}(\cdot|s) - \pi_k(\cdot|s), p(\cdot|s) - \pi_{k+1}(\cdot|s) \rangle$$
$$= \|p(\cdot|s) - \pi_k(\cdot|s)\|^2 - \|\pi_{k+1}(\cdot|s) - \pi_k(\cdot|s)\|^2 - \|p(\cdot|s) - \pi_{k+1}(\cdot|s)\|^2. \tag{31}$$

Combining (30) and (31), we have that Lemma 4.4 holds.

## E  Proof of Lemma 4.5

Plug $p = \pi_k$ in (14), we have that

$$\eta_k \langle Q_{\mathcal{P}}^{\pi_k}(s, \cdot), \pi_{k+1}(\cdot|s) - \pi_k(\cdot|s) \rangle \leq -\|\pi_{k+1}(\cdot|s) - \pi_k(\cdot|s)\|^2 - \|\pi_k(\cdot|s) - \pi_{k+1}(\cdot|s)\|^2. \tag{32}$$

Moreover, plug $p = \pi^*$ in (14), we have that

$$\eta_k \langle Q_{\mathcal{P}}^{\pi_k}(s, \cdot), \pi_{k+1}(\cdot|s) - \pi_k(\cdot|s) \rangle + \eta_k \langle Q_{\mathcal{P}}^{\pi_k}(s, \cdot), \pi_k(\cdot|s) - \pi^*(\cdot|s) \rangle$$
$$\leq -\|\pi_{k+1}(\cdot|s) - \pi_k(\cdot|s)\|^2 + \|\pi^*(\cdot|s) - \pi_k(\cdot|s)\|^2 - \|\pi^*(\cdot|s) - \pi_{k+1}(\cdot|s)\|^2. \tag{33}$$

For the term $\langle Q_{\mathcal{P}}^{\pi_k}(s, \cdot), \pi_{k+1}(\cdot|s) - \pi_k(\cdot|s) \rangle$, we have that

$$g_{\mathcal{P}}^{\pi_{k+1}} - g_{\mathcal{P}}^{\pi_k} \overset{(a)}{\leq} \mathbb{E}_{s \sim d_{\mathsf{P}_{\pi_{k+1}}}^{\pi_k}} \left[ \langle Q_{\mathcal{P}}^{\pi_k}(s, \cdot), \pi_{k+1}(\cdot|s) - \pi_k(\cdot|s) \rangle \right]$$

$$\overset{(b)}{\leq} \frac{1}{M} \mathbb{E}_{s \sim d_{\mathsf{P}_{\pi_k}}^{\pi^*}} \left[ \langle Q_{\mathcal{P}}^{\pi_k}(s, \cdot), \pi_{k+1}(\cdot|s) - \pi_k(\cdot|s) \rangle \right] \leq 0, \tag{34}$$

where $(a)$ is from the performance difference lemma [8] and $(b)$ is from the definition of $M$ and (32). For the term $\langle Q_{\mathcal{P}}^{\pi_k}(s, \cdot), \pi_k(\cdot|s) - \pi^*(\cdot|s) \rangle$, we have that

$$0 \leq g_{\mathcal{P}}^{\pi_k} - g_{\mathcal{P}}^* \leq g_{\mathsf{P}_{\pi_k}}^{\pi_k} - g_{\mathsf{P}_{\pi_k}}^* = \mathbb{E}_{s \sim d_{\mathsf{P}_{\pi_k}}^{\pi^*}} \left[ \langle Q_{\mathcal{P}}^{\pi_k}(s, \cdot), \pi_k(\cdot|s) - \pi^*(\cdot|s) \rangle \right]. \tag{35}$$

Therefore, we have that

$$M \left( g_{\mathcal{P}}^{\pi_{k+1}} - g_{\mathcal{P}}^{\pi_k} \right) + g_{\mathcal{P}}^{\pi_k} - g_{\mathcal{P}}^*$$
$$\leq \mathbb{E}_{s \sim d_{\mathsf{P}_{\pi_k}}^{\pi^*}} \left[ \frac{1}{\eta_k} \|\pi^*(\cdot|s) - \pi_k(\cdot|s)\|^2 \right] - \mathbb{E}_{s \sim d_{\mathsf{P}_{\pi_k}}^{\pi^*}} \left[ \frac{1}{\eta_k} \|\pi^*(\cdot|s) - \pi_{k+1}(\cdot|s)\|^2 \right]. \tag{36}$$

By re-arranging the terms in the above equation, we have that

$$g_{\mathcal{P}}^{\pi_{k+1}} - g_{\mathcal{P}}^* \leq \frac{M-1}{M} (g_{\mathcal{P}}^{\pi_k} - g_{\mathcal{P}}^*) + \frac{1}{M} \mathbb{E}_{s \sim d_{\mathsf{P}_{\pi_k}}^{\pi^*}} \left[ \frac{1}{\eta_k} \|\pi^*(\cdot|s) - \pi_k(\cdot|s)\|^2 \right]$$
$$- \frac{1}{M} \mathbb{E}_{s \sim d_{\mathsf{P}_{\pi_k}}^{\pi^*}} \left[ \frac{1}{\eta_k} \|\pi^*(\cdot|s) - \pi_{k+1}(\cdot|s)\|^2 \right]. \tag{37}$$

# F Proof of Theorem 4.6

By recursively applying Lemma 4.5, we have that

$$g_{\mathcal{P}}^{\pi_k} - g_{\mathcal{P}}^* \leq \left(1 - \frac{1}{M}\right)^k \left(g_{\mathcal{P}}^{\pi_0} - g_{\mathcal{P}}^*\right) + \left(1 - \frac{1}{M}\right)^{k-1} \mathbb{E}_{s \sim d_{\mathsf{P}_{\pi_0}}^{\pi^*}} \left[\frac{1}{\eta_0}\left\|\pi^*(\cdot|s) - \pi_0(\cdot|s)\right\|^2\right]$$

$$+ \frac{1}{M} \sum_{t=1}^{k-1} \left(\left(1 - \frac{1}{M}\right)^{k-t-1} \mathbb{E}_{s \sim d_{\mathsf{P}_{\pi_t}}^{\pi^*}} \left[\frac{1}{\eta_t}\left\|\pi^*(\cdot|s) - \pi_t(\cdot|s)\right\|^2\right]\right.$$

$$\left. - \left(1 - \frac{1}{M}\right)^{k-t} \mathbb{E}_{s \sim d_{\mathsf{P}_{\pi_{t-1}}}^{\pi^*}} \left[\frac{1}{\eta_{t-1}}\left\|\pi^*(\cdot|s) - \pi_t(\cdot|s)\right\|^2\right]\right). \tag{38}$$

Since the step size satisfies $\eta_k \geq \eta_{k-1}\left(1 - \frac{1}{M}\right)^{-1} M$, we have that

$$\frac{1}{M} \sum_{t=1}^{k-1} \left(\left(1 - \frac{1}{M}\right)^{k-t-1} \mathbb{E}_{s \sim d_{\mathsf{P}_{\pi_t}}^{\pi^*}} \left[\frac{1}{\eta_t}\left\|\pi^*(\cdot|s) - \pi_t(\cdot|s)\right\|^2\right]\right.$$

$$\left. - \left(1 - \frac{1}{M}\right)^{k-t} \mathbb{E}_{s \sim d_{\mathsf{P}_{\pi_{t-1}}}^{\pi^*}} \left[\frac{1}{\eta_{t-1}}\left\|\pi^*(\cdot|s) - \pi_t(\cdot|s)\right\|^2\right]\right) \leq 0. \tag{39}$$

Therefore, we have that

$$g_{\mathcal{P}}^{\pi_k} - g_{\mathcal{P}}^* \leq \left(1 - \frac{1}{M}\right)^k \left(g_{\mathcal{P}}^{\pi_0} - g_{\mathcal{P}}^*\right) + \left(1 - \frac{1}{M}\right)^{k-1} \frac{1}{M\eta_0} \mathbb{E}_{s \sim d_{\mathsf{P}_{\pi_0}}^{\pi^*}} \left[\left\|\pi^*(\cdot|s) - \pi_0(\cdot|s)\right\|^2\right]. \tag{40}$$

# G Proof of Theorem 4.7

Set the step size $\eta_k = \frac{1}{L}$ and define the gradient mapping $G_L(\pi_k(\cdot|s)) = L(\pi_k(\cdot|s) - \pi_{k+1}(\cdot|s))$. Note that if $\pi_k$ is updated exactly by the (sub)-gradient descent, then $G_L(\pi_k(\cdot|s)) = \nabla g_{\mathcal{P}}^\pi(s, \cdot)$. The norm $\|G_L(\pi_k(\cdot|s))\|$ measures the closeness of the current step to the first-order stationary point. To prove Theorem 4.7, we first show the following lemma.

**Lemma G.1.** *For the robust average cost, setting $L = L_\pi$ we have the following inequalities.*

$$\left(\frac{g_{\mathcal{P}}^{\pi_{k+1}} - g_{\mathcal{P}}^*}{C_{PL} 2\sqrt{2|S|}}\right)^2 \leq L\left(g_{\mathcal{P}}^{\pi_k} - g_{\mathcal{P}}^{\pi_{k+1}}\right) \tag{41}$$

*Proof.* For the update of our algorithm

$$\pi_{k+1}(\cdot|s) = \arg\min_{p \in \Delta(\mathcal{A})} \left\{\eta\langle Q_{\mathcal{P}}^{\pi_k}(s, \cdot), p\rangle + D(p, \pi_k(\cdot|s))\right\}, \forall s \in \mathcal{S}, \tag{42}$$

the following first-order optimality condition holds for any $p \in \Pi$ and $s$

$$\left\langle Q_{\mathcal{P}}^{\pi_k}(s, \cdot) + L\left(\pi_{k+1}(\cdot|s) - \pi_k(\cdot|s)\right), p(\cdot|s) - \pi_{k+1}(\cdot|s)\right\rangle \geq 0. \tag{43}$$

By letting $p = \pi_k$, we have that

$$\left\langle Q_{\mathcal{P}}^{\pi_k}(s, \cdot), \pi_k(\cdot|s) - \pi_{k+1}(\cdot|s)\right\rangle$$

$$\geq \left\langle L\left(\pi_{k+1}(\cdot|s) - \pi_k(\cdot|s)\right), \pi_{k+1}(\cdot|s) - \pi_k(\cdot|s)\right\rangle$$

$$= L\|\pi_{k+1}(\cdot|s) - \pi_k(\cdot|s)\|^2$$

$$= \frac{1}{L}\|G_L(\pi_k(\cdot|s))\|^2 \tag{44}$$

From Lemma 4.1, we further have that

$$g_{\mathcal{P}}^{\pi_k} - g_{\mathcal{P}}^{\pi_{k+1}} \geq \mathbb{E}_{s \sim d_{\mathsf{P}_{\pi_{k+1}}}^{\pi_{k+1}}} \left[\left\langle Q_{\mathsf{P}_{\pi_k}}^{\pi_k}(s, \cdot), \pi_k(\cdot|s) - \pi_{k+1}(\cdot|s)\right\rangle\right]. \tag{45}$$

Therefore, we have that

$$g_{\mathcal{P}}^{\pi_k} - g_{\mathcal{P}}^{\pi_{k+1}} \geq \mathbb{E}_{s \sim d_{\mathsf{P}_{\pi_{k+1}}}^{\pi_{k+1}}} \left[\frac{1}{L}\|G_L(\pi_k(\cdot|s))\|^2\right]. \tag{46}$$

On the other hand, from Lemma 4.2, we have that

$$g_{\mathcal{P}}^{\pi_{k+1}} - g_{\mathcal{P}}^* \le C_{PL} \max_{\hat{\pi}} \mathbb{E}_{s \sim d_{\mathsf{P}_{\pi_{k+1}}}^{\pi_{k+1}}} \left[ \langle Q_{\mathcal{P}}^{\pi_{k+1}}(s, \cdot), \pi_{k+1}(\cdot|s) - \hat{\pi}(\cdot|s) \rangle \right]. \tag{47}$$

For $\langle Q_{\mathcal{P}}^{\pi_{k+1}}(s, \cdot), \pi_{k+1}(\cdot|s) - \hat{\pi}(\cdot|s) \rangle$, from the first-order optimality condition (43), we have that

$$\langle Q_{\mathcal{P}}^{\pi_{k+1}}(s, \cdot), \pi_{k+1}(\cdot|s) - \hat{\pi}(\cdot|s) \rangle$$
$$\le \langle Q_{\mathcal{P}}^{\pi_{k+1}}(s, \cdot), \pi_{k+1}(\cdot|s) - \hat{\pi}(\cdot|s) \rangle + \langle Q_{\mathcal{P}}^{\pi_k}(s, \cdot) + L(\pi_{k+1}(\cdot|s) - \pi_k(\cdot|s)), p(\cdot|s) - \pi_{k+1}(\cdot|s) \rangle$$
$$= \langle Q_{\mathcal{P}}^{\pi_{k+1}}(s, \cdot) - Q_{\mathcal{P}}^{\pi_k}(s, \cdot), \pi_{k+1}(\cdot|s) - \hat{\pi}(\cdot|s) \rangle - \langle L(\pi_{k+1}(\cdot|s) - \pi_k(\cdot|s)), \pi_{k+1}(\cdot|s) - \hat{\pi}(\cdot|s) \rangle$$
$$\overset{a}{\le} (L_\pi + L)\|\pi_{k+1}(\cdot|s) - \pi_k(\cdot|s)\|\|\pi_{k+1}(\cdot|s) - \hat{\pi}(\cdot|s)\|$$
$$\overset{(b)}{\le} (L_\pi + L)\|\pi_{k+1}(\cdot|s) - \pi_k(\cdot|s)\|\sqrt{2|S|}$$
$$\le \left(1 + \frac{L_\pi}{L}\right)\|G_L(\pi_k(\cdot|s))\|\sqrt{2|S|}, \tag{48}$$

where $(a)$ is due to the fact that the robust relative action value function $Q_{\mathcal{P}}^\pi$ is $L_\pi$-Lipschitz (Lemma 4.3) and (b) is due to the fact that $\|\pi_{k+1}(\cdot|s) - \hat{\pi}(\cdot|s)\| \le \sqrt{2|S|}$.

Therefore, we have that

$$g_{\mathcal{P}}^{\pi_{k+1}} - g_{\mathcal{P}}^* \le C_{PL} \max_{\hat{\pi}} \mathbb{E}_{s \sim d_{\mathsf{P}_{\pi_{k+1}}}^{\pi_{k+1}}} \left[ \left(1 + \frac{L_\pi}{L}\right)\|G_L(\pi_k(\cdot|s))\|\sqrt{2|S|} \right]. \tag{49}$$

By setting $L = L_\pi$, we have that

$$\left( \frac{g_{\mathcal{P}}^{\pi_{k+1}} - g_{\mathcal{P}}^*}{C_{PL} 2\sqrt{2|S|}} \right)^2 \le \left( \mathbb{E}_{s \sim d_{\mathsf{P}_{\pi_{k+1}}}^{\pi_{k+1}}} \left[ \|G_{L_\pi}(\pi_k(\cdot|s))\| \right] \right)^2$$
$$\le \mathbb{E}_{s \sim d_{\mathsf{P}_{\pi_{k+1}}}^{\pi_{k+1}}} \left[ \|G_{L_\pi}(\pi_k(\cdot|s))\|^2 \right] \le L_\pi \left( g_{\mathcal{P}}^{\pi_k} - g_{\mathcal{P}}^{\pi_{k+1}} \right), \tag{50}$$

where the second inequality is from Jensen's inequality. $\qquad\square$

We are now ready to prove Theorem 4.7.

*Proof.* Let $\omega = \frac{1}{(C_{PL} 2\sqrt{2|S|})^2}$. We have that

$$0 \le \frac{\omega}{L_\pi}(g_{\mathcal{P}}^{\pi_{k+1}} - g_{\mathcal{P}}^*)^2 \le g_{\mathcal{P}}^{\pi_k} - g_{\mathcal{P}}^{\pi_{k+1}}. \tag{51}$$

Let $\delta_k = g_{\mathcal{P}}^{\pi_k} - g_{\mathcal{P}}^*$. We then have that

$$\frac{\omega}{L_\pi}\delta_{k+1}^2 \le (g_{\mathcal{P}}^{\pi_k} - g_{\mathcal{P}}^*) + (g_{\mathcal{P}}^* - g_{\mathcal{P}}^{\pi_{k+1}}) = \delta_k - \delta_{k+1}. \tag{52}$$

We then divide both side by $\delta_k \delta_{k+1}$ and have that

$$\frac{1}{\delta_{k+1}} - \frac{1}{\delta_k} \ge \frac{\omega}{L_\pi} \frac{\delta_{k+1}}{\delta_k}. \tag{53}$$

We then take the sum of the above inequality over iterations $0, 1, \cdots, k-1$. It then follows that

$$\frac{1}{\delta_k} - \frac{1}{\delta_0} \ge \frac{\omega}{L_\pi} \sum_{i=0}^{k-1} \frac{\delta_{i+1}}{\delta_i}. \tag{54}$$

From (46), we have that $g_{\mathcal{P}}^{\pi_k} - g_{\mathcal{P}}^{\pi_{k+1}} \ge \mathbb{E}_{s \sim d_{\mathsf{P}_{\pi_{k+1}}}^{\pi_{k+1}}} \left[ \frac{1}{L_\pi}\|G_{L_\pi}(\pi_k(\cdot|s))\|^2 \right] \ge 0$. Therefore, we have that $g_{\mathcal{P}}^{\pi_k} \ge g_{\mathcal{P}}^{\pi_{k+1}}$ and thus $\delta_{i+1} \le \delta_i$.

For any two constant $l, m \in (0, 1)$, let $n(k, l)$ be the number of steps that $\frac{\delta_{i+1}}{\delta_i}$ is at least $l$ over the first $k$ iterations. If $n(k, l) \ge mk$, we then have that $\frac{\delta_{i+1}}{\delta_i} \ge l$ at least $\lceil mk \rceil$ times. Therefore,

$$\frac{1}{\delta_k} - \frac{1}{\delta_0} \ge \frac{\omega}{L_\pi} mlk. \tag{55}$$

We then have that

$$\delta_k \leq \frac{L_\pi}{\omega m l k}.$$  (56)

If $n(k, l) < mk$, we have that $\frac{\delta_{i+1}}{\delta_i} < l$ at least $\lceil (1 - m)k \rceil$ times. Since $\delta_{i+1} \leq \delta_i$, we have that

$$\delta_k \leq \delta_0 l^{(1-m)k} = \delta_0 (l^{(1-m)})^k.$$  (57)

Since (56) and (57) hold for any $l, m \in (0, 1)$, we have that

$$\delta_k \leq \min_{0 < l, m < 1} \max \left\{ \frac{L_\pi}{\omega m l k}, \delta_0 (l^{(1-m)})^k \right\}.$$  (58)

By letting $l = m = \frac{1}{2}$, we have that

$$g_{\mathcal{P}}^{\pi_k} - g_{\mathcal{P}}^* \leq \max \left\{ \frac{4L_\pi}{\omega k}, \left( \frac{\sqrt{2}}{2} \right)^k \left( g_{\mathcal{P}}^{\pi_0} - g_{\mathcal{P}}^* \right) \right\}.$$  (59)

This completes the proof. $\qquad\square$

