# OpenReview forum: "Policy Optimization for Robust Average Reward MDPs"
_NeurIPS.cc/2024/Conference — NeurIPS 2024 poster_

### Official Review · Reviewer_2EU7 · 2024-07-08

**Soundness:** 3
**Presentation:** 3
**Contribution:** 3
**Rating:** 7
**Confidence:** 4

**Summary:**

The authors introduce study a policy gradient algorithm for solving unichain average reward robust MDPs. They show a linear convergence rate for increasing step sizes and a $O(1/k)$ convergence rate for fixed step size, where $k$ is the number of iterations of the algorithms.

**Strengths:**

This is a good paper. It is well-written and defines clearly its objectives. It extends to the case of unichain average reward robust MDPs the current state-of-the-art results for policy gradient methods in discounted robust MDPs and average reward nominal MDPs.

**Weaknesses:**

I only have three minor comments as regards the claims of the authors, which I find a bit too far from the exact results proved in their theorems. I think that the paper requires some clarifications.

* The authors should mention the unichain assumption much earlier in the paper. The word *unichain* isn’t even present in the abstract, in the introduction, it is only mentioned in the very end of section 2. This assumption is crucial to establish several important properties of average value function (such as uniformity across states) and relative value function (such as continuity). This needs to be emphasized much earlier, and highlighted when the authors give their main contributions. Given the connection between mean-payoff perfect-information stochastic games and robust MDPs [0], the sentence from l13-14 of the abstract: *Moreover, our algorithm is the first algorithm that converges to the global optimum with polynomial time for robust average reward MDPs* would mean that the authors have solved a long-standing open problem, which is not the case since they solely focus on unichain instances (although I appreciate the results of the authors). Please clarify.

* I am a bit confused with the claim about the polynomial time complexity of the algorithms. Usually, a (weakly) polynomial time complexity refers to a number of arithmetic iterations polynomial in $\log(1/\epsilon)$, in the dimension of the problems and in the logarithm of the entries of the problem (here, the number of bits necessary to represent the rewards and the transition probabilities). So why is $O(1/\epsilon)$ a polynomial-time complexity? Similarly, the $O(.)$ notations seem to be hiding the $C_{PL}$ term from (15), and it’s not clear how this is bounded in terms of $S,A$ and the parameter of the problem. Same with the constant $L_{\pi}$ from (16), can you provide a bound on this? The closed statement from polynomial-time complexity appears to be Th 4.6 but there again we have terms like $M$ appearing in the bound, and it’s not clear how to bound them. Please clarify or reformulate your claims about *polynomial-time* algorithms.

* Some of the definitions appear incorrect. In all generality the definition of the bias should involve some Cesaro averages or the sum may diverge, the definition given by the authors is correct only for aperiodic Markov chains (see my other comments below).

[0] Chatterjee, K., Goharshady, E. K., Karrabi, M., Novotný, P., & Žikelić, Đ. (2023). Solving long-run average reward robust mdps via stochastic games. arXiv preprint arXiv:2312.13912.

**Questions:**

* The authors in [1] mention a $O(1/\epsilon)$ convergence rate for their algorithm in their conclusion. How does that compare to your $O(1/\epsilon)$ result?

* Under the unichain assumption, it is well-known that discounted MDP with a discount factor sufficiently large can solve average reward MDPs (typically with a discount factor larger than $1-\epsilon/D$ where $D$ is the diameter of the MDP). There is a vast number of recent papers on policy gradient methods for nominal discounted MDPs and discounted robust MDPs. Can the authors compare the complexity of their algorithms with the complexity of the best policy gradient methods for discounted MDPs combined with the appropriate (large) discount factor for unichain MDPs?

* I understand that the robust value function may be non-smooth for robust MDPs. But isn’t it also the case in the discounted setting? How do authors in the discounted setting bypass that for designing policy gradient algorithms? Is it the same `trick' in your paper? Also, for what geometry of the uncertainty set is the robust value function smooth?

* l159-160, definition of the bias: in all generality the definition of the bias/relative value function should involve the Cesaro average, see p 338 in [2]. In my understanding the definition given by the authors only holds fors for *aperiodic* Markov chains. Please clarify.

* A minor comment: Usually, rewards are maximized, and costs are minimized. In this paper, rewards are minimized, which is a bit weird, even though the maths don’t change.

* Please introduce the equations for $d^{\pi}$ in the main body.

l287: typo, the first two sentences of rmk 4.8 should be one sentence.

[1] A. Tewari and P. L. Bartlett. Bounded parameter markov decision processes with average reward criterion. In International Conference on Computational Learning Theory, pages 263–277. Springer, 2007.

[2] M. L. Puterman. Markov decision processes: Discrete stochastic dynamic programming, 1994

**Limitations:**

No problem here.

---

> ### Author Rebuttal · Authors · 2024-08-06
>
> We thank the Reviewer for the helpful and insightful feedback. Below we provide point-to-point responses to the weaknesses and questions.
>
> **W1. Present assumption 2.1 earlier.** In the revision, we will introduce the unichain assumption earlier in abstract and emphasize that our contributions are based on the unichain assumption.
>
> **W2. Polynomial time complexity of the algorithm.**
> Since the complexity of our algorithm is in the order of $\mathcal{O}(1/\epsilon)$ not $\mathcal{O}(\log(1/\epsilon))$, the polynomial-time complexity claim is not accurate. We will reformulate our claims in the revision. Though our algorithm doesn't have polynomial-time complexity, our algorithm is still the first algorithm that has finite time complexity for robust average reward MDPs.
>
> Since $M$ and $C_{PL}$ depend on the structure of the underlying MDPs, characterizing the upper bound of $M$ and $C_{PL}$ is non-trivial. We would like to emphasize that similar constants also appear in prior works on robust discounted reward MDPs and average reward MDPs ([3], [4], [5], [6]). For the constant $L_\pi$, when the underlying MDP is ergodic, the upper bound of $L_\pi$ is characterized in [6], which is also depends on the structure of the MDP. From the Lemma 14 in [6], we have that $L_\pi$ is in the order of $\mathcal{O}\Big(\frac{C_e^2S^2}{(1-\lambda)^2}\sqrt{A}\Big)$, where $C_e$ and $\lambda$ are constants that characterize the geometric ergodicity of the MDPs.
>
> **W3. Clarify definitions of average reward.**
> In [2], the Cesaro limit sense relative value function was defined for periodic chain. This is because the Markov chain is not assumed to be unichain in [2]. When the Markov chain is unichain, the Markov chain only contains a single recurrent class and some transient states, and the average reward is independent of the initial state. Since there is only one recurrent class, the relative value function does not need to involve the Cesaro average.
>
> **Q1. Compare with an existing work with $\mathcal{O}(1/\epsilon)$ convergence rate.**
> In [1], the $\mathcal{O}(1/\epsilon)$ convergence rate was achieved for robust average reward MDPs when the uncertainty set is a bounded parameter MDP. Specifically, for each state action pair $(s, a)$, the transition probability $p_s^a(s^\prime)$ is bounded between a lower bound $l_s^a(s^\prime)$ and an upper bound $u_s^a(s^\prime)$. In our paper, we consider more general uncertainty sets. Moreover, [1] studies the convergence rate by building the connection between discounted reward MDPs and average reward MDPs while our paper focuses on policy gradient-based method. We discussed [1] in the related works section in our paper. We will also compare the $\mathcal{O}(1/\epsilon)$ convergence rate in the revision.
>
> **Q2. Compare with policy gradient method for discounted MDPs with large discount factor.**
> There are recently works on policy gradient methods for robust discounted MDPs. In [3], the robust MDPs are considered under the $R$-contamination uncertainty set. The robust policy gradient converges to the global optimum with iteration complexity $\mathcal{O}(1/\epsilon^3)$. In [4], the double-loop robust policy gradient was proposed and was further proved to converge to the global optimum with complexity $\mathcal{O}(1/\epsilon^4)$. In [6], with sufficiently large step size, the robust policy mirror descent converges to the global optimum with complexity  $\mathcal{O}(1/\epsilon)$. To the best of the author's knowledge, there is no works building the connection between the robust discounted reward and robust average reward. Therefore, it's non-trivial to compare the complexity of our algorithm with the policy gradient methods for discounted MDPs combined with the large discount factor.
>
> The authors of [7] shows that there exists a discount factor
> $\gamma_{bw}\in(0, 1)$ such that any policy that is $\gamma$-discount optimal for $\gamma\in(\gamma_{bw}, 1)$ is also Blackwell optimal when the uncertainty set is definable. However, how to choose $\gamma_{bw}$ is still an open problem. In [7], only value iteration based algorithms are proposed and no stopping criterion is introduced for their algorithms.
>
> **Q3. Bypass non-smoothness of robust value function in discounted MDPs.**
> For discounted reward MDPs, different methods are proposed to bypass the non-smoothness of the robust value function. In [3], the total-variation uncertainty set is studied and the non-smoothed robust value function is smoothed by approximating the max operator using the LogSumExp (LSE) operator. In [4], the non-smoothness of the robust value function is bypassed by introducing the Moreau envelope function, which can also be viewed as a smoothed version of the robust value function. In [5], the authors develop a similar bound as the Lemma 4.1 in our paper by applying the Bellman equation of robust value function, and further apply it to show the convergence of the mirror descent algorithm. However, it is not applicable in our case as the average reward doesn't satisfy the Bellman equation.
>
>  A sufficient condition for the robust value function to be smooth is that there exists a constant $C$ such that $\|d\_s\^{\pi, \mathsf{P}\_{\pi}} - d\_s\^{\pi\^\prime, \mathsf{P}\_{\pi\^\prime}}\|\leq C\\|\pi-\pi\^\prime\\|$, which depends on the geometry of the uncertainty set. Since it involves the worst-case transition kernel and the closed-form expression doesn't exist, it unclear for what geometry of the uncertainty set is the robust value function smooth.
>
> **Q4. Clarify the definition of relative value function.**
> Please refer to the response of W3
>
> **Q5. Average cost.**
> In the revision, we will replace rewards with costs.
>
> **Q6. Introduce the equations for $d^\pi$.**
> We provide the equations for $d^\pi_\mathsf{P}$ in the revision: $d^\pi_\mathsf{P}$ is the stationary measure of $s$ under the transition kernel $\mathsf{P}$ satisfies that $d^\pi_\mathsf{P} \mathsf{P} = d^\pi_\mathsf{P}$.
>
> **Q7. Typo.**
> Fixed.

---

> > ### Comment · Reviewer_2EU7 · 2024-08-10
> > **Response to the authors**
> >
> > I would like to thank the authors for their (very!) detailed response to my comments. Provided that they indeed make all the changes listed above, the paper will read better and contribute to the literature on gradient methods for robust MDPs. I increased my score in consequence.

---

> > > ### Author Response · Authors · 2024-08-10
> > >
> > > We thank the reviewer again for the response and helpful comments, which greatly helped to improve the quality of the
> > > paper.

---

### Official Review · Reviewer_H19Z · 2024-07-11

**Soundness:** 3
**Presentation:** 3
**Contribution:** 3
**Rating:** 6
**Confidence:** 4

**Summary:**

The paper studies gradient-based methods for robust average reward MDPs. The paper first derives a sub-gradient for the robust average reward (which is nonsmooth), and then uses it to define a mirror descent algorithm. They prove a few structural properties of the setting and then use them to provide a convergence guarantee for their algorithm. They also provide some experimental results.

**Strengths:**

Overall the paper and proofs are written with high clarity and quality, but I do have several technical concerns (in the questions section). I would raise my soundness score and overall evaluation if they were all adequately addressed.

There is good originality in the problem setting and the fact that they consider gradient-based methods. In particular it requires surmounting certain technical difficulties related to the robust average reward. If the technical concerns can be addressed, then I think these theoretical developments could prove to have good significance to future work.

**Weaknesses:**

The main weakness of the paper is what I perceive to be several technical concerns (listed in the questions section). I would raise my soundness score and overall evaluation if they were all adequately addressed.

**Questions:**

Maybe the unichain assumption 2.1 should be placed earlier, since the stationary distributions $d^\pi$ are not generally well-defined in the absence of this assumption.

In line 177, is the definition of $d_{\mathcal{P}}^\pi$ unique?

Lemma 3.2: similar question, I think the worst-case transition kernel for a policy $\pi$ is not unique. In this case I wonder if $d_{\mathcal{P}}^\pi$ and $Q_{\mathcal{P}}^\pi$ are uniquely defined? And thus my main question is, for the sub-gradient, can we use different worst-case transition kernels in $d_{\mathcal{P}}^\pi$ and $Q_{\mathcal{P}}^\pi$, or do they need to use the same one? (The proof makes it seem like they are using the same one?) Thus maybe the lemma statement would need to be amended to discuss this.

Lemma 4.2: Is $C_{PL}$ guaranteed to be finite? The usual definition of unichain (e.g. in Puterman) is that the transition matrix has a single recurrent class plus a (possibly empty) set of transient states. The transient states would have stationary measure 0 which could cause $C_{PL}$ to blow up?

I have a similar concern about the quantity $M$ in Lemma 4.5. I am also confused about the denominator (is there a missing subscript $\pi$?)

Where is the proof of Lemma 4.3? It is claimed that a result from [16] can be easily extended to a more general (unichain) setting; however, because such details can be very thorny in the average reward setting, I think a proof should be provided.

**Limitations:**

No major limitations

---

> ### Author Rebuttal · Authors · 2024-08-06
>
> **W1. Listed in the question section.**
> Please refer to the response of Q1-Q6.
>
> **Q1. Unichain assumption should be placed earlier.**
> We thank the reviewer for this comment. In the revision, we will introduce the unichain assumption earlier in the paper and emphasize that our contributions are based on the unichain assumption.
>
> **Q2. Uniqueness of $d_\mathcal{P}^\pi$.**
> The worst-case transition kernel may not be unique for general uncertainty sets, and thus $d_\mathcal{P}^\pi$ may not be unique.
>
> **Q3. Uniqueness of $d_\mathcal{P}^\pi$ and $Q_\mathcal{P}^\pi$.**
> Though the worst-case transition kernel may not be unique, $Q\_\mathcal{P}\^\pi$ is unique. From the definition of the worst-case transition kernel, for a worst-case transition kernel $\mathsf{P}\_\pi$ of a policy $\pi$, we have that $Q\_{\mathsf{P}\_\pi}\^\pi = \max\_{\mathsf{P}\in\mathcal{P}} Q\_{\mathsf{P}}\^\pi$. Therefore, the robust action value function is unique. For the sub-gradient, the worst-case transition kernel of $d\_\mathcal{P}\^\pi$ and $Q\_\mathcal{P}\^\pi$ can be different, but this does not affect our results.
>
> **Q4. Boundedness of $C_{PL}$.**
> We agree with the reviewer that in the unichain setting, $C_{PL}$ might not be finite if transient state exists.
> The constant $C_{PL}$ depends on the structure of the underlying MDPs and also appears in prior works on discounted reward MDPs and non-robust average reward MDPs ([1], [2], [3], [4]). If the Markov chain is ergodic, i.e., there is only a single recurrent class, then $C_{PL}$ is guaranteed to be finite.
>
> **Q5. Boundedness of $M$.**
> Similar as $C_{PL}$, the constant $C_{PL}$ depends on the structure of the underlying MDPs. If the Markov chain is ergodic, then $M$ is guaranteed to be finite. The definition of $M$ is our paper is accurate. For the denominator, the subscript $\mathsf{P}$ denotes the transition kernel that we maximize. In the numerator, the subscript $\mathsf{P}_\pi$ denotes the worst-case transition kernel of $\pi$.
>
> **Q6. Proof of Lemma 4.4.**
> In Lemma 14 of [4], the $L_\pi$-Lipschitz of the relative value function is proved. In their paper, $L_\pi = 2C_m^2C_p\kappa_r + 2C_mC_r$. For $C_p$ and $\kappa_r$, the proof of Lemma 18 in [4] shows that the ergodicity of the Markov chain is not required and the results can be extended to the unichain setting. We only need to show that $C_m$ is finite for unchain. In [4], $C_m$ is defined as the maximum of the operator norm of the matrix $(I-\Phi P_\pi)^{-1}$ across all policies $\pi\in\Pi$. For any $\pi\in\Pi$, following the proofs in Section A.2 of [4], we have that the eigenvalues of $(I-\Phi P_\pi)$ is non-zero for unchain since unchain only has a single recurrent class and the stationary distribution exists. Therefore, for any $\pi\in\Pi$, $\|(I-\Phi P_\pi)^{-1}\| < \infty$. Moreover, since the collection of all policy $\Pi$ is compact, we have that the maximum of the operator norm of the matrix $(I-\Phi P_\pi)^{-1}$ across all policies $\pi\in\Pi$ is finite. Therefore, $L_\pi$ exists for unchains. We will add the proof sketch of Lemma 4.4 in the revision.
>
> **Reference.**
> [1] Y. Wang and S. Zou. Policy gradient method for robust reinforcement learning. In Proc.
> International Conference on Machine Learning (ICML), volume 162, pages 23484–23526. PMLR, 2022.
>
> [2] Q. Wang, C. P. Ho, and M. Petrik. Policy gradient in robust mdps with global convergence415
> guarantee, 2023.
>
> [3] Y. Li, G. Lan, and T. Zhao. First-order policy optimization for robust markov decision process.
> arXiv preprint arXiv:2209.10579, 2022.
>
> [4] N. Kumar, Y. Murthy, I. Shufaro, K. Y. Levy, R. Srikant, and S. Mannor. On the global convergence of policy gradient in average reward markov decision processes. arXiv preprint arXiv:2403.06806, 2024.

---

> > ### Comment · Reviewer_H19Z · 2024-08-10
> >
> > Thank you for your response. From this response it seems like therefore the results are based on the ergodic (one recurrent class with no transient states) assumption, rather than unichain? If so, I would be happy to increase my score if this is presented earlier and more centrally in the paper.

---

> > > ### Author Response · Authors · 2024-08-10
> > >
> > > We thank the reviewer for the response. After reading the reviewer's comments, we found that assuming ergodicity instead of unchain is more reasonable. In this case, we don't need to make additional assumptions on $C_{PL}$ and $M$. We will present the assumption earlier in the paper and make it clear.

---

### Official Review · Reviewer_7myA · 2024-07-12

**Soundness:** 3
**Presentation:** 3
**Contribution:** 3
**Rating:** 6
**Confidence:** 3

**Summary:**

The authors present a gradient-based algorithm for average reward robust MDP (finite MDP). This setting has been studied in prior works, however, this paper proposes a policy optimization-based algorithm which is not yet done. They do a theoretical analysis of this setting and show linear convergence (by increasing step size), finite iteration complexity and quite nicely show satisfying PL condition and therefore perhaps trivially global convergence.

**Strengths:**

- The paper is easy to read.
- Policy gradient-based approach for robust average rewards MDP
- There are multiple theoretical contributions: Linear convergence, Global optimality, derived sub-gradient of robust average reward MDPs

**Weaknesses:**

- Presentation of theoretical results is not sharp or rather quite bad. They do not mention the assumptions for the lemma/theorem statement holds. I am uncertain if there are any further assumptions.
-  Much of the used theory is already exists in the literature of robust MDP and infinite horizon case (which is not bad since authors make sure to explain that they not trivially using but extending it, e.g., 197-204 lines). This is currently described in pieces throughout the paper, I would appreciate it if authors could concretely explain at one place (perhaps in related works) the tools used from prior works.

**Questions:**

Questions:

43-44 -> What are these algorithms based upon in contrast to gradient-based?

Is there an assumption on eqn 3 about boundedness? Why does V and Q are necessarily finite?

What will be a sufficient condition for assumption 2.1? Also motivating beforehand on why you need this assumption will be good.

Is there a bound/relation between C_{PL} and M?

What does the non-robust method line 307 optimize for? what is it's objective and used transition?

---

> ### Author Rebuttal · Authors · 2024-08-06
>
> We thank the Reviewer for the helpful and insightful feedback. Below we provide point-to-point responses to the weaknesses and questions.
>
> **W1. Didn't mention the assumptions for the lemma/theorem statements.**
> We thank the reviewer for this comment. In the revision, we will rephrase the statements of lemmas and theorems to include the assumptions we use. Specifically,
> Theorem 4.6: Under Assumption 2.1, set the step size $\eta_k \geq \eta_{k-1}\Big(1-\frac{1}{M}\Big)^{-1}M$, the robust policy mirror descent satisfies
> $g\_\mathcal{P}\^{\pi\_k} - g\_\mathcal{P}\^\* \leq (1-\frac{1}{M})\^k(g\_\mathcal{P}^{\pi\_0} - g\_\mathcal{P}\^\*) + (1-\frac{1}{M})\^{k-1}\frac{1}{M\eta\_0}\mathbb{E}\_{s\sim d\_{\mathsf{P}\_{\pi\_0}}\^{\pi\^\*}}[\|\pi\^\*(\cdot|s) - \pi\_0(\cdot|s)\|\^2].$
>
> For Theorem 4.7:  Under Assumption 2.1, let step size $\eta = \frac{1}{L_\pi}$ for all $k\geq 1$, we have that the each iteration of the robust policy mirror descent satisfies $g\_\mathcal{P}\^{\pi\_k} - g\_\mathcal{P}\^\* \leq \max\Big\\{\frac{4L_\pi}{\omega k},(\frac{\sqrt{2}}{2})^k\big(g_\mathcal{P}^{\pi_0} - g_\mathcal{P}^*\big)\Big\\}$,
> where $\omega = (2\sqrt{2S}C_{PL})^{-2}$.
>
> **W2. Explain the tools used from prior works at one place.**
> We thank the reviewer for this comment. We summarize the results that are used from prior works in Section 1.2. We will add the following paragraph in the revision:
>
> In [1], the Fr\'echet sub-gradient has been derived for robust discounted reward MDPs. However, the Fr\'echet sub-gradient for robust discounted reward MDPs in [1] can not be extended to the average setting since in [1], the discounted factor $\gamma$ is required to be strictly less than 1. In [2], the performance difference lemme was derived for average reward MDPs. In our paper, we provide a lower bound on the difference of robust average reward between two policies. Such a bound was also derived in [1] for robust discounted value function by applying the Bellman equation of robust value function, which is not applicable in our case as the average reward itself doesn’t satisfy the Bellman equation. We also extend the Lipschitz property of the non-robust relative value function in [3] to the unichain setting. In [4] and [5], the convergence rate of the policy gradient descent for non-robust discounted reward MDPs was derived. In their analyses, the smoothness of the value function is required. However, the robust average reward might not be smooth. Therefore, the approaches applied in [4] and [5] can not be directly extended to robust average reward MDPs.
>
> **Q1. Details on related works.**
> [6] and [10] build the connection between the discounted reward MDPs and average reward MDPs and prove the existence of Blackwell optimal policies. The algorithms based on value iteration are further proposed. [7] is an extension of [6] with parts of the state space having arbitrary transitions and other parts are purely stochastic. In [8] and [9], the model-based and model-free robust average reward MDPs are studied and the robust relative value iteration algorithms are proposed. We will add this discussion in the revision
>
> **Q2. Boundedness assumption on V and Q.**
> We thank the reviewer for pointing this out. Since $V$ and $Q$ are unique up to an additive constant, similar as [3], we consider the projection of the value function onto the subspace orthogonal to the $\textbf{1}$ vector. In this case, our theoretical results still hold.
>
> **Q3. Sufficient condition and motivation for assumption 2.1.**
> The sufficient condition for Assumption 2.1 is that for any $\pi\in\Pi$ and $\mathsf{P}\in \mathcal{P}$, the Markov chain only contains a single recurrent class and some transient states.  The unichian assumption is important and widely used since under the unichain assumption, the robust average reward is identical for every starting state. To derive Lemma 4.1, we leverage the fact that the robust average reward is independent of the initial state. If the unichain assumption doesn't hold, the inequality may not hold, and this is a key step to derive the global optimality of our algorithm.
>
> In practice, it is often the case that only unichains are of interest to the decision-making problem. For example, the nominal transition kernel is obtained from samples, and is a unichain. Then, the true transition kernel must be a unichain, and therefore to obtain a robust policy that works well on the true transition kernel, it is sufficient to only consider unichain. Even in standard MDP, extending results from unichain to e.g., multichain problems is very challenging. Extending our results under a relaxed assumption beyond unichain is even more challenging, and it is of future interest.
>
> **Q4. Relation between $C_{PL}$ and $M$.**
> The relation between $C_{PL}$ and $M$ is unclear. We have that $M = \sup\_{\pi, \mathsf{P}\in\mathcal{P}}\Big\\|\frac{d\_{\mathsf{P}\_\pi}\^{\pi\^\*}}{d\_\mathsf{P}\^\pi}\Big\\|\_\infty = \sup\_{\pi, s, \mathsf{P}\in\mathcal{P}} \frac{d\_{\mathsf{P}\_\pi}\^{\pi^*}(s)}{d\_\mathsf{P}\^\pi(s)}$ and $C\_{PL} = \max\_{\pi, s} \frac{d\_{\mathcal{P}}\^{\pi\^\*}(s)}{d\_\mathcal{P}\^\pi(s)} = \max\_{\pi, s} \frac{d\_{\mathsf{P}\_{\pi\^\*}}\^{\pi\^\*}(s)}{d\_{\mathsf{P}\_{\pi}}\^\pi(s)}$. The sup of $M$ and $C_{PL}$ can be achieved at different $\pi$. In the numerator, it's unclear if $d\_{\mathsf{P}\_\pi}\^{\pi\^\*}(s)$ is larger than $d\_{\mathsf{P}\_{\pi\^\*}}\^{\pi\^\*}(s)$ or not.
>
> **Q5. Objective and transition kernel of baseline algorithm.**
> For the baseline algorithm, it's trained under the known nominal transition kernel and the objective is to maximize the non-robust average reward.

---

> ### Comment · Reviewer_7myA · 2024-08-10
>
> I thank the authors for their replies. I went over them and do not have any further questions. Please include the changes as promised in your revised version, particularly regarding assumptions.

---

> > ### Author Response · Authors · 2024-08-10
> >
> > We thank the reviewer for the response. We will revise our paper based on the comments.

---

### Official Review · Reviewer_Ehp7 · 2024-07-14

**Soundness:** 3
**Presentation:** 3
**Contribution:** 3
**Rating:** 6
**Confidence:** 5

**Summary:**

The authors consider the mirror descent algorithm in the context of robust average cost MDPs. The consider $(s,a)$-rectangular uncertainty sets across a general distance metric. The Bregman divergence chosen for the purpose of analysis is the Euclidean 2-norm distance. The authors leverage on a prior result proving the existence of a corresponding Bellman equation for the robust average cost and another prior result proving Lipschitzness of the risk neutral average cost value function to derive the results in the paper. Since the robust average cost is the maximum average cost across the uncertainty set it is not differentiable and hence the authors resort to using the sub-gradient as a substitute. This sub gradient is shown to be the robust $Q$ function. Subsequently, the performance difference inequality in conjunction with a PL result is proven and leveraged to obtain the final convergence bounds which are of $O(1/\epsilon)$. They characterize these bounds for both a constant and increasing step size.

**Strengths:**

1. The problem considered is of significance to the community. Average cost problems are harder to analyze than their discounted cost counterparts due to the absence of the contraction factor. Hence even though its a technical challenge, this objective is more representative of applications where long term performance is important. And the robust average cost objective is important to ensure optimality in the face of uncertainty associated with the underlying transition kernel.

2. Even though the analysis techniques are inspired by prior literature such as Agarwal et al, Xiao, it is not a straight forward extension and hence this paper provides with valuable tools in this context such as the performance difference inequality, PL inequality, etc which may be of independent interest to the community.

3. The paper yields $O(1/\epsilon)$ convergence bounds which are optimal even when compared to the risk neutral average cost setting/discounted cost setting.

4. The results are also experimentally validated.

**Weaknesses:**

1. The authors seem to have missed a line of work pertaining to the robust average cost. Although they consider general $(s,a)$-rectangular sets with any distance metric, much of the prior work has considered the KL (Kullback Leibler) distance metric whose dual formulation yields the exponential cost robust MDPs. Some of the earlier works on this include [Borkar 2010], [BS02] where they characterize asymptotic behavior of dynamic programming algorithms for this robust formulation. Some of the more recent work in this domain can be found in [MMS23] and [MMS24]. [MMS23] considers a policy gradient algorithm for this robust setting, but provides asymptotic convergence bounds to a stationary point. [MMS24] considers modified policy iteration, an algorithm that generalizes both policy and value iteration and provides with finite time global convergence guarantees. Hence the repeated claim in the paper to have characterized the first policy based global convergence bounds in the context of robust average cost MDPs has to be modified accordingly.

2. The authors assume an oracle provides with $Q^\pi_\mathcal{P}$, that is the value function corresponding to the argmax kernel associated with policy $\pi$. In the risk neutral case, one can solve for the state action value function provided corresponding to a policy $\pi$ by solving for the average cost Bellman equation (which typically involves a matrix inversion). It is not clear as to how to evaluate this robust value function even when you have access to the model, that is the nominal transition matrix and the single step costs.

3. The authors assume that the underlying Markov chain is unichain. However, unichain Markov chain implies the existence of some transient states whose associated stationary measure is zero. In that case, the concentrability coefficient $C_{PL}$ will be infinity. Hence it seems like ergodicity is a requirement.

4. The increasing step size requires knowledge of $M$, which is not apparent even in the full information setting. For instance, reference [16] in the paper characterizes the Lipschitz constant in terms of the model considered and hence can be evaluated when given access to the complete model information such as the probability transition matrix, etc. However, it is not apparent as to how to determine M based on its definition in line 254.



[Borkar 2010] Borkar, V. S. (2010, July). Learning algorithms for risk-sensitive control. In Proceedings of the 19th International Symposium on Mathematical Theory of Networks and Systems–MTNS (Vol. 5, No. 9).

[BS02] Borkar, V. S., & Meyn, S. P. (2002). Risk-sensitive optimal control for Markov decision processes with monotone cost. Mathematics of Operations Research, 27(1), 192-209.

[MMS24] Moharrami, M., Murthy, Y., Roy, A., & Srikant, R. (2024). A policy gradient algorithm for the risk-sensitive exponential cost MDP. Mathematics of Operations Research.

[MMS23] Murthy, Y., Moharrami, M., & Srikant, R. (2023, June). Modified Policy Iteration for Exponential Cost Risk Sensitive MDPs. In Learning for Dynamics and Control Conference (pp. 395-406). PMLR.

**Questions:**

1. Is there a closed form expression for the value function corresponding to the solution of Equation 7? In exponential cost MDPs, these value functions do indeed have a closed form expression and I was curious about their generalizations to other distance metrics.

2. Since the Bregman divergence considered is the Euclidean 2 norm, it feels more accurate to refer to the algorithm as policy gradient rather than mirror descent. Mirror descent is more commonly employed when the KL divergence is used as the Bregman divergence. Is there any specific reason for not referring to this algorithm as policy gradient?

3. Since the minimization problem is considered, it is more relevant to refer to the setting as average cost rather than average reward as is currently done.


Some typos:

1. Line 463.5 in the appendix: should be $+\sum_a\pi(a|s)$ instead of $-\sum_a\pi(a|s)$

2. In equation 10: should be $\nabla g^{\pi_k}_\mathcal{P}$ instead of $\nabla g^{\pi}_\mathcal{P}$

**Limitations:**

Even though the limitations are not explicitly addressed, the paper has been presented clearly and hence the limitations are apparent.

---

> ### Author Rebuttal · Authors · 2024-08-06
>
> We thank the Reviewer for the helpful and insightful feedback. Below we provide point-to-point responses to the weaknesses and questions.
>
> **W1. Related works on robust average cost MDPs.**
> We thank the reviewer for pointing this out. In the revision, we modify our statement as 'We characterized the first policy based global convergence bounds with general uncertainty sets for robust average cost MDPs.' We also compare our work with the related works [Borkar 2010], [BS02], [MMS24], [MMS23]. We add the following paragraph in Section 1.2 in the revision:
>
> **Exponential cost robust MDPs.** For the robust average cost MDPs, when the uncertainty set is defined by the KL-divergence metric, the problem admits a dual formulation, which is the exponential cost robust MDPs. The exponential cost robust MDPs have also been studied in the literature. In [Borkar 2010], the Q-learning and the actor-critic method are described and the asymptotic performance are characterized for risk sensitive robust MDPs. In [BS02], the value iteration and policy iteration algorithms are also analyzed for risk sensitive MDPs. Recently, in [MMS23], the modified policy iteration is proved to converge to the global optimum for exponential cost risk sensitive MDPs. The policy gradient algorithm for the risk sensitive exponential cost MDPs is studied and the asymptotic convergence bounds to a stationary point are provided in [MMS24]. In our paper, we study the robust average reward MDPs with general uncertainty sets and characterize the global convergence of our algorithm.
>
> **W2. Evaluate the robust relative value function.**
> The robust relative value function can be evaluated by generalizing the relative value iteration to the robust setting. The algorithm is presented as follows:
>
> > Algorithm parameters: $V_0$, $\epsilon$, and reference state $s^* \in \mathcal{S}$
> Initialize  $w_0 \leftarrow V_0 - V_0(s^*)\textbf{1}$
> >
> > Loop for each step: $sp(w_t - w_{t-1})\geq \epsilon$
> $\quad$Loop  for  each  $s \in \mathcal{S}$:
> $\qquad V\_{t+1} \leftarrow \mathbb{E}\_{a\sim\pi}[r(s, a) + \sigma\_{\mathcal{P}\_s\^a}(w\_t)]$
> $\qquad w\_{t+1}(s) \leftarrow V\_{t+1}(s) - V\_{t+1}(s^*)$
> return $V_t$
>
> In the algorithm, $\\sigma\_{\\mathcal{P}\_s\^a}(w\_t) = \\max_{p\\in\\mathcal{P}\_s\^a}p\\top w\_t$, and can be solved in the dual formulation for various uncertainty sets [Iyengar05]. The returned $V\_t$ is the robust relative value function and $V\_t(s\^*)$ is the robust average reward.
>
> The assumption that an oracle that outputs the robust value function exists is also presented in robust discounted reward MDPs, for example [Li22].
>
> **W3. Unichain assumption may lead to infinite $C_{PL}$.**
> We agree with the reviewer that the unichain assumption may not guarantee that $C_{PL}$ is finite, and moreover, ergodicity is a sufficient condition to guarantee $C_{PL}$ is finite.
>
> **W4. Determine the increasing step size $M$.**
> Since the definition of $M$ involves taking the sup over all transition kernels in the uncertainty set, theoretically determining $M$ is challenging. In practice, we only require an upper bound for $M$ so that the convergence result holds in Theorem 4.6. Therefore, in practice, we can run simulation to find an upper bound of $M$ and fine tune it. For example, we can numerically solve $\min_{\pi, P\in\mathcal{P}}\\|d_P^\pi\\|_\infty$ when the ergodicity holds, which is an upper bound of $M$.
>
> **Q1. Closed from expression for the value function.**
> For the widely used metrics like the total variation distance, $\chi$ -square distance and the KL divergence, the closed-form expression exists for $\max\_{\mathsf{P}\in\mathcal{P}}\sum\_{s\^\prime \in \mathcal{S}}\mathsf{P}\_{s, s\^\prime}\^aV\_{\mathcal{P}}\^\pi(s\^\prime)$ [Iyengar05]. The dual formulation depends on the structure of the uncertainty set. However, if the value function has closed-form solution as the exponential cost MDPs is unclear. Moreover, our algorithm and theoretical results do not depend on the specific structure of the uncertainty set and the closed-form solution of the value function.
>
>
> **Q2. Policy gradient or mirror descent.**
> We don't refer to our algorithm as policy gradient is due to the fact that in our algorithm, we replace the policy (sub)-gradient $\nabla g_\mathcal{P}^{\pi_k}$ by $Q_\mathcal{P}^{\pi_k}$. Since our algorithm is the mirror descent with a specific Bregman divergence and $Q_\mathcal{P}^{\pi_k}$ is not the (sub)-gradient, we refer to our algorithm as policy mirror descent.
>
> **Q3. Average cost MDPs.**
> We thank the reviewer for this comment. In our paper, we adopt a minimization formulation to align with conventions in the optimization literature. In the revision, we will replace rewards with costs.
>
> **Q4. Typos.** We thank the reviewer for pointing this out. Fixed.
>
> **Reference.**
> [Borkar 2010] Borkar, V. S. (2010, July). Learning algorithms for risk-sensitive control. In Proceedings of the 19th International Symposium on Mathematical Theory of Networks and Systems–MTNS (Vol. 5, No. 9).
>
> [BS02] Borkar, V. S., Meyn, S. P. (2002). Risk-sensitive optimal control for Markov decision processes with monotone cost. Mathematics of Operations Research, 27(1), 192-209.
>
> [MMS24] Moharrami, M., Murthy, Y., Roy, A., Srikant, R. (2024). A policy gradient algorithm for the risk-sensitive exponential cost MDP. Mathematics of Operations Research.
>
> [MMS23] Murthy, Y., Moharrami, M., Srikant, R. (2023, June). Modified Policy Iteration for Exponential Cost Risk Sensitive MDPs. In Learning for Dynamics and Control Conference (pp. 395-406). PMLR.
>
> [Iyengar05] Garud N. Iyengar (2005). Robust Dynamic Programming. Mathematics of Operations Research.
>
> [Li22] Yan Li, Guanghui Lan, Tuo Zhao. (2022). First-order Policy Optimization for Robust Markov Decision Process.

---

> > ### Comment · Reviewer_Ehp7 · 2024-08-11
> >
> > Thanks for addressing my comments!
> >
> > I am satisfied with the responses and am increasing my score. I am still not convinced that the algorithm should be referred to as policy mirror descent since the Euclidean norm as Bregman divergence yields policy gradient. I understand the need to use the sub gradient (due the maximization objective), but the divergence considered is however the Euclidean norm and hence policy gradient (or sub-gradient)  is perhaps more accurate than mirror descent.
> >
> > Once again, thank you for your response!

---

### Decision · Program_Chairs · 2024-09-25

**Decision:**

Accept (poster)

**Comment:**

The paper analyzed policy gradient methods for average reward robust MDPs.
The contribution and challenge here are mainly technical, since the results for the discounted case already appear in previous work.
Nevertheless, all reviewers (some of them experts in the field) unanimously agree that the contribution is important and relevant to the neurips community and that the paper is well written.
I will therefore accept.